# Tackling Provably Hard Representative Selection via Graph Neural Networks

**Mehran Kazemi**                                    *mehrankazemi@google.com*
**Anton Tsitsulin**                                  *tsitsulin@google.com*
**Hossein Esfandiari**                               *esfandiari@google.com*
**MohammadHossein Bateni**                           *bateni@google.com*
**Deepak Ramachandran**                              *ramachandrand@google.com*
**Bryan Perozzi**                                    *bperozzi@acm.org*
**Vahab Mirrokni**                                   *mirrokni@google.com*
*Google Research*

**Reviewed on OpenReview:** *https://openreview.net/forum?id=3LzgOQ3eOb*

## Abstract

*Representative Selection* (RS) is the problem of finding a small subset of exemplars from a dataset that is representative of the dataset. In this paper, we study RS for attributed graphs, and focus on finding representative nodes that optimize the accuracy of a model trained on the selected representatives. Theoretically, we establish a new hardness result for RS (in the absence of a graph structure) by proving that a particular, highly practical variant of it (RS for Learning) is hard to approximate in polynomial time within any reasonable factor, which implies a significant potential gap between the optimum solution of widely-used surrogate functions and the actual accuracy of the model. We then study the setting where a (homophilous) graph structure is available, or can be constructed, between the data points. We show that with an appropriate modeling approach, the presence of such a structure can turn a hard RS (for learning) problem into one that can be effectively solved. To this end, we develop *RS-GNN*, a representation learning-based **RS** model based on **G**raph **N**eural **N**etworks. Empirically, we demonstrate the effectiveness of RS-GNN on problems with predefined graph structures as well as problems with graphs induced from node feature similarities, by showing that RS-GNN achieves significant improvements over established baselines on a suite of eight benchmarks.

## 1 Introduction

In the age of massive data, having access to tools that can select exemplar data points representative of an entire dataset is of crucial importance. *Representative selection* (RS) (Feldman, 2020), finding a small subset of exemplars from an unlabeled dataset that transmits maximal information for a certain objective, has numerous applications in summarization, active learning, data compression, model training cost reduction, and many other domains (see, e.g., (Bairi et al., 2015; Kaushal et al., 2019; Liu et al., 2015; Wei et al., 2014; You et al., 2020; Church & ReVelle, 1974; Killamsetty et al., 2021c; Feldman, 2010)).

In this work, we study RS for attributed graphs. We first study the computational complexity of a specific but widely-applicable formulation of the RS problem in the absence of a graph structure, where we attempt to find a fixed-size subset of representative exemplars from a dataset that can be used to train a model with the best possible *accuracy* on the entire dataset. We show it is impossible to provide a polynomial-time RS algorithm with an approximation factor better than $\omega(n^{-1/\text{poly} \log \log n})$, unless the Exponential Time Hypothesis (ETH) fails. ETH is a widely-believed assumption in the domain of parameterized complexity which states that the 3-SAT problem cannot be solved in subexponential time in the worst case. Note that $\omega(n^{-1/\text{poly} \log \log n})$ is almost polynomial, ruling out the existence of any constant approximation or even poly-logarithmic approximation.

Our subconstant hardness result is of particular importance because several previous works find representatives by optimizing *surrogate functions*—these can be approximated well in theory—instead of the actual model accuracy. For instance, they consider a submodular surrogate function which can be approximated within a factor $1 - 1/e$ in polynomial time (Guillory, 2012; Wei et al., 2015; Mirzasoleiman et al., 2020; Chen & Krause, 2013; Esfandiari et al., 2021). Our hardness result implies that, in the worst case, there is a significant gap between the optimum solution of such surrogate functions and the actual accuracy of the model, rendering the surrogate functions poor estimators for the quality of the model. To the best of our knowledge, this is the first subconstant hardness result for the RS problem. This motivates us to deviate from directly defining proxy functions, and make use of learning-based approaches that discover the hidden structure of the data to guide the selection. This is in line with the recent attempts to solve computationally hard problems with neural networks, e.g., (Wilder et al., 2019; Coleman et al., 2019).

We next study the setting where besides the node features, a (homophilous) graph structure between the data points is provided, which can help guide the selection process. We show empirically that with an appropriate modeling approach, the presence of such a graph structure can turn an originally hard RS problem into one that can be effectively solved. To this end, we develop RS-GNN: a learning-based model for **R**epresentatives **S**election via **G**raph **N**eural **N**etworks. We first demonstrate the effectiveness of RS-GNN for selecting representative nodes from datasets where a natural graph can be accessed, i.e., where edges may be specified by some natural property of the data (e.g., paper citations). Then, we demonstrate that even when a natural graph is not available but a similarity graph is believed to have some degree of homophily, creating a similarity graph of the input data points and applying RS-GNN can still select high-quality representatives. We conduct experiments on eight datasets with different sizes and properties, and in both settings where we have and do not have access to a graph structure. Our results show that our model provides significant improvements over three kinds of baselines: 1) well-established baselines that optimize predefined surrogate functions, 2) learning-based methods utilizing graph clustering/pooling and 3) baselines based on active learning.

Our main contributions are: 1) Providing a hardness result establishing that, under a standard computational-complexity assumption, RS is hard to approximate in polynomial time within any reasonable factor (this is the *first* subconstant hardness result for RS, to the best of our knowledge), 2) Demonstrating empirically that the existence of a homophilous graph structure between the data points can make hard RS problems effectively solvable, and 3) Developing RS-GNN for effective RS when one has access to such a graph structure and showing its merit for datasets with natural and/or similarity graphs.

## 2   Related Work

We group the main existing work from the literature that relates to our paper as follows. Further discussion and other categories of related work can be found in Appendix F.

**Active learning:** In active learning (Settles, 2009; Cohn et al., 1996) we have an unlabeled set of data points that we can request to label. Since labeling is an expensive task, we usually have a limited budget, say, we can label up to $k$ data points, which are then used to predict the labels of all data points. The goal is to select the set of data points to label in such a way as to maximize the accuracy of the final model. The data points can be iteratively selected in mini-batches (select a mini-batch, label the data points in the batch, update the model, and repeat), or in one-shot (Hoi et al., 2006; Guo & Schuurmans, 2007; Chakraborty et al., 2014; Citovsky et al., 2021; Amin et al., 2020). The latter is typically used when model training is time-consuming. RS can be used in the context of one-shot active learning, or for selecting the first mini-batch in the context of mini-batch active learning. In these contexts, a common approach to active learning is to use unsupervised surrogate functions such as KMediod (Schubert & Rousseeuw, 2019) and MaxCover (Hochbaum & Pathria, 1998) to select a set of data points that maximally cover the dataset with respect to some objective. Moreover, active learning models have been developed for attributed graphs both for mini-batched labeling (Cai et al., 2017; Gao et al., 2018) and for one-shot (Wu et al., 2019b; Zhang et al., 2021). We compare against many surrogate functions as well as one-shot graph active learning models in our experiments.

**Hardness of Clustering:** The hardness problem we study is distantly related to clustering, which has come in many flavors and shapes: flat vs. hierarchical, partitioning vs. overlapping, graph-based vs. embedding-based vs. time-series-based, supervised vs unsupervised, etc. The interested reader may refer to references

(e.g., Hastie et al. (2009); Kaufman & Rousseeuw (2009); Gan et al. (2007); Xu & Wunsch (2005); Jain & Dubes (1988); Xu & Tian (2015); Ezugwu et al. (2022); Silwal et al. (2023)) for further information. We do emphasize here, though, that the most similar clustering objectives to what we study here are the center-based clustering problems such as k-center and k-means. In these settings, the hardness results and known algorithmic guarantees (i.e., lower and upper bounds) are not far from each other. For example, while non-metric $k$-center cannot be approximated to within any constant, the metric special case (generalizing the ubiquitous Euclidean setting) admits a 2-approximation (Gonzalez, 1985) and cannot be approximated to within better than a factor 2 (Vazirani, 2013). On the other hand, the $k$-means objective admits a constant-factor approximation (Ahmadian et al., 2020; Kanungo et al., 2002) and the best hardness results are 1.0013 (Awasthi et al., 2015; Lee et al., 2017). In contrast to all these results, we present a superconstant hardness for the RS problem, stressing the big gap between any optimizable surrogate function and the true objective.

**Graph clustering (community detection):** Early approaches only considered the graph structure and disregarded the node features. These approaches typically learn an embedding for each node (e.g., the spectral features, random walk embeddings, or auto-encoder based embeddings) and then feed these node embeddings into a clustering algorithm such as kmeans (see, e.g., Cao et al. (2015); Nikolentzos et al. (2017); Grover & Leskovec (2016); Ye et al. (2018)). Recently, approaches based on GNNs, which take both graph structure and node features into account, have gained more popularity and success (Zhang et al., 2019; Peng et al., 2021; Liu et al., 2022b; Wang et al., 2019; Bo et al., 2020; Kamhoua et al., 2022; Tsitsulin et al., 2023). Graph pooling and joint clustered representation learning approaches are also relevant (Liu et al., 2022a; Ding et al., 2021; Van Den Oord et al., 2017). RS and clustering are two highly related tasks: many models developed for RS can (in theory) be used for clustering and vice versa. However, due to the distinct properties of the two tasks, a model that works well for one task may not necessarily work well for the other. While our main focus is on RS, we also experiment with graph clustering and compare against several existing approaches. We also experiment with the relevant works from the graph pooling literature both for RS and clustering.

## 3 Notation & Problem Definition

We use bold lowercase letters to denote vectors and bold uppercase letters to denote matrices. Let $\boldsymbol{x}_i$ represent the $i^{\text{th}}$ element of $\boldsymbol{x}$ and $\boldsymbol{M}_i$ represent the $i^{\text{th}}$ row of $\boldsymbol{M}$. For a function $f : A \mapsto B$ and a subset $A' \subseteq A$, we use $f|_{A'}$ to denote the restriction of the domain of $f$ to $A'$. For a dataset, we use $m$ to represent the number of data points and $n$ to represent the number of features. We use $\mathcal{V} = \{v_1, \ldots, v_m\}$ to represent the set of data points and $\boldsymbol{X} \in \mathbb{R}^{m \times n}$ to represent the feature matrix, such that $\boldsymbol{X}_i$ corresponds to the features of $v_i$. When data points have class labels, we use $c$ to represent the number of classes. First, we define a general framework for Representative Selection (RS) as follows:

**Definition 1** (RS)**.** *Given a set of data points $\mathcal{V}$, their features $\boldsymbol{X}$, a number $0 < k \leq |\mathcal{V}|$, and a utility function $u : 2^{\mathcal{V}} \mapsto \mathbb{R}$, the representative selection problem is to select a subset $\mathcal{S} \subseteq \mathcal{V}$ of $k$ representatives that maximize the utility $u(\mathcal{S})$.*[1]

Note that the number of features $n$ is unrelated to the number of data points $m$ and the number of selected points $k$. The applicability and tractability of an RS problem depends on the utility function, $u$. Intuitively, $u$ should capture the usefulness of the subset $\mathcal{S}$ as a representative of $\mathcal{V}$; more precisely, if there is a particular application of the full dataset $\mathcal{V}$, $u$ quantifies the degree to which $\mathcal{S}$ can be used instead. This can vary with the particular application considered; in this paper we are mostly concerned with a particular utility model that associates representativeness with *learnability*. In the *Representative Selection for Learning* (RSL) problem, we get to see the labels of the selected representatives, based on which we aim to train a classifier with highest predictive performance on the entire dataset.

**Definition 2** (RSL)**.** *Let $\mathcal{V}$ be a set of data points with observed features $\boldsymbol{X}$ and with labels $Y$ generated from an oracle function $\phi^* : \mathcal{V} \mapsto \{1, \ldots, c\}$, and $\Phi$ be a class of predictor functions (not necessarily containing $\phi^*$). Given $\mathcal{V}$, $\boldsymbol{X}$, $\Phi$, $\phi^*$, and a number $0 < k \leq |\mathcal{V}|$, the goal of RSL is to select a subset $\mathcal{S}$ of $k$ representatives from $\mathcal{V}$ such that training a classifier $\phi \in \Phi$ based on $\boldsymbol{X}$ and $Y|_{\mathcal{S}}$ (obtained by querying $\phi^*$) maximizes the normalized accuracy of $\phi$ on the entire set $\mathcal{V}$.*

---

[1]Note that $\mathcal{V}$ is implicit in the definition of $u$, hence in the definition of RS.

The model class $\Phi$ defines a suitable inductive hypothesis for the learning problem so that RS is well-defined. The predictor $\phi$ is chosen to maximize the normalized accuracy, defined as $\overline{\text{Acc}} = c(\text{accuracy} - 1/c)$, on the entire dataset; in other words it is a *transductive learning problem* (Gammerman et al., 1998). RSL is a natural problem that has multiple real-world applications in dataset selection, active learning, efficient ML and other areas (see section 7 for examples). Further, it can be expected to correspond to a general notion of the representativeness of a subset, even outside a learning use case. In the rest of this paper we will focus on this version of the RS problem, and defer generalizations to other utility models to future work.

We now define Graph RSL (GRSL), a version of RSL for attributed graphs. We denote an attributed graph as $\mathcal{G} = \{\mathcal{V}, \boldsymbol{A}, \boldsymbol{X}\}$ where $\mathcal{V} = \{v_1, \ldots, v_m\}$ represents the set of nodes, $\boldsymbol{A} \in \mathbb{R}^{m \times m}$ represents the adjacency matrix, and $\boldsymbol{X} \in \mathbb{R}^{m \times n}$ represents the matrix of node features ($m$ nodes and $n$ features).

**Definition 3** (GRSL)**.** *Let $\mathcal{G} = \{\mathcal{V}, \boldsymbol{A}, \boldsymbol{X}\}$ be an attributed graph with node labels $Y$ generated from an oracle function $\phi^* : \mathcal{V} \mapsto \{1, \ldots, c\}$, and $\Phi$ be a class of predictor functions (not necessarily containing $\phi^*$). Given $\mathcal{G}$, $\Phi$, $\phi^*$, and a number $0 < k \le |\mathcal{V}|$, the goal of GRSL is to select a subset $\mathcal{S}$ of $k$ representatives from $\mathcal{V}$ such that training a classifier $\phi \in \Phi$ based on $\boldsymbol{X}$, $\boldsymbol{A}$, and $Y|_{\mathcal{S}}$ (obtained by querying $\phi^*$) maximizes the normalized accuracy of $\phi$ on the entire set $\mathcal{V}$.*

## 4 Theoretical Findings: Hardness Results for RSL

We start by analyzing the hardness of the RSL problem. Our worst case results for RSL extend to GRSL because in the worst case, the graph can be random and provide no extra information.

We describe a theoretical finding that explains why the common practice of hand designing and optimizing surrogate functions may not be a good approach for RSL. In Definition 3, let $u(S) = \overline{\text{Acc}}(\phi(S))$ represent the normalized accuracy of the classifier $\phi$ when trained on a subset $\mathcal{S}$ of data points. Let $\mathcal{S}^* = \arg\max_{\mathcal{S}} u(\mathcal{S})$ be the optimal set of representatives. Ideally, one would optimize $u(\mathcal{S})$ and find $\mathcal{S}^*$. This may, however, be impossible without a-priori having access to the labels for all data points. Alternatively, most existing works on RSL (and RS in general) focus on defining an intuitive surrogate function $\Omega$ and find a solution $\mathcal{S}^\Omega$ by optimizing $\Omega(\mathcal{S})$ in the hope that $u(\mathcal{S}^\Omega)$ is a good approximator for $u(\mathcal{S}^*)$. In this section, we establish an inapproximability result demonstrating that $u(\mathcal{S}^\Omega)$ may not be a good estimator of $u(\mathcal{S}^*)$ for any polynomial-time-computable surrogate function $\Omega$. We do this by showing that there are naturally defined learning tasks for which there exists a significant gap between $u(\mathcal{S}^\Omega)$ and $u(\mathcal{S}^*)$.

We start by studying the computational hardness of RSL on an end-to-end binary classification task, from which the aforementioned claims follow. We say an RS algorithm $\mathcal{A}$ (e.g., optimizing a surrogate function) approximates the optimal solution with an approximation factor $\alpha$ if we can establish that $u(\mathcal{S}^{\mathcal{A}})$ is within a multiplicative factor $\alpha$ of the optimal solution $u(\mathcal{S}^*)$ for any learning problem, where $\mathcal{S}^{\mathcal{A}}$ is the output of the RS algorithm.

Under the Exponential-Time Hypothesis (ETH) assumption, we show that there is no polynomial-time RS algorithm with an approximation factor better than $\omega(n^{-1/\text{poly} \log \log n})$. Exponential-Time Hypothesis is an assumption that is widely used in the domain of fixed-parameter tractability to prove hardness results[2]. This assumption has also been used to show strong approximation gaps (Manurangsi (2017); Huchette et al. (2020)). The Exponential-Time Hypothesis is defined as follows.

**Definition 4** (Exponential-Time Hypothesis)**.** *There exists a positive real number $\delta$ such that 3-SAT can not be solved in $O(2^{\delta n} \text{poly}(n))$ time, where $n$ is the number of parameters in $k$-SAT.*

This is a stronger assumption than P$\neq$NP. In simple words, P$\neq$NP states that 3-SAT cannot be solved in polynomial time, while ETH states that 3-SAT cannot be solved in subexponential time.

Here, we show that there is an instance of RSL for which the gap between $u(\mathcal{S}^{\mathcal{A}})$ and $u(\mathcal{S}^*)$ for any polynomial-time RSL algorithm $\mathcal{A}$ is at least $\omega(n^{-1/\text{poly} \log \log n})$, unless ETH fails. A similar approximation gap exists between the best polynomial-time and best exponential-time algorithm. Note that $\omega(n^{-1/\text{poly} \log \log n})$ is almost polynomial, ruling out the existence of any constant approximation or even poly-logarithmic approximation.

---

[2]E.g., see Aggarwal & Stephens-Davidowitz (2018); Braverman et al. (2017; 2014); Chen et al. (2012; 2021); Cygan et al. (2015); Jonsson et al. (2013); Lokshtanov et al. (2011))

Also note that the best solution may not give 100% accuracy, nor does it necessarily match the accuracy obtained by using all labels (since we only use $k$).

**Definition 5** (Fit-or-Not (FoN) Learning Problem)**.** *We have $m$ data points and $n$ binary features. Each feature is associated with one of two types: red or blue. The types are generated independently and uniformly at random, they are consistent across data points, and are hidden from the algorithm. Each data point has value 1 for two features and 0 for the rest. The label of a data point is 1 if the type of its features of value 1 are the same, and the label is 0 otherwise. The goal is to maximize the normalized accuracy for all the data points given labels only on selected data points.*

Figure 1 visually presents an instance of the FoN problem. Each row represents a data point and each column represents a feature. There are $m = 10$ data points, each having $n = 5$ binary features. Each data point has a value of 1 exactly for two of the features, and a value of 0 for the rest. The types of the first and the third features are *blue* and the types of the other three features are *red*. These types are hidden from the algorithm. For the first data point, the two values of 1 are for the first and second features. Since these features have different types, the label for this data point is 0. For the second data point, however, the two values of 1 are for the first and the third features that have the same type, therefore the label for this data point is 1. The labels of the other data points are determined similarly.

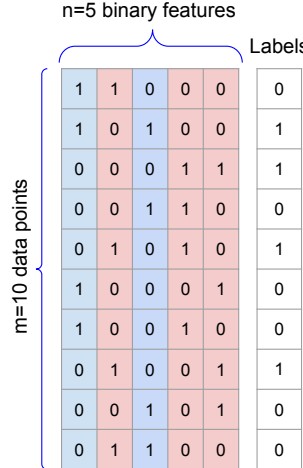

Figure 1: An instance of the FoN problem with $m = 10$ data points and $n = 5$ binary features.

The goal of the algorithm is to select a subset with $k < m$ data points in such a way that a model trained on the labels of those $k$ data points makes accurate predictions for all the $m$ data points (i.e. it generalizes well to the other $(m - k)$ data points).

FoN can be naturally framed as RSL by defining the components from Definition 3 as follows: let $\mathcal{V}$ be the set of $m$ data points, $\boldsymbol{X} \in \mathbb{R}^{m \times n}$ be the features matrix, $\Phi$ be the class of models that correspond to the data generation process in Definition 5 (constructed from particular partitions of features into red/blue etc.), $k$ be the budget for selecting representatives, and let the labels $\phi^*$ be as defined above; the latent information consists of the types of the $n$ features. The simplicity of FoN shows that RS is hard in a very broad form.

Note that FoN is a basic artificial learning problem that we defined to show that *there exists* a hard RSL problem, and hence theoretically RSL in its most general form can not be solved in polynomial time. However, we can provide some semi-natural interpretation of this problem. Consider the following example. We have $m$ users (corresponding to points) and $n$ political news articles (corresponding to features). Each article is generated by either a conservative perspective or liberal perspective. However, we are *not aware* of the perspective of each article. Note that each user selects two article as their favorite. Each user can be represented by a binary vector of features of length $n$, where their favorite articles are marked by 1. We say a user is neutral if she likes two articles of different perspective (i.e. one conservative and one liberal). Otherwise, we say the user is supportive. The task of FoN is to learn whether a user is supportive or neutral.

The next theorem is the main result of this section.

**Theorem 1.** *There is no polynomial-time RSL algorithm for FoN with an approximation factor better than $\omega(n^{-1/\operatorname{poly} \log \log n})$, unless the exponential-time hypothesis fails.*

**Proof Sketch (See full proof in Appendix C)**     *The proof proceeds by reducing the densest $k$-subgraph problem to the FoN problem. In the densest $k$-subgraph problem, the goal is to find a subgraph with $k$ vertices and the maximum number of edges for an unweighted graph. We say an algorithm is an $\alpha$-approximation algorithm for the densest $k$-subgraph problem if it returns a subgraph with $k$ vertices where the number of edges is at least $\alpha$ times that of the densest $k$-subgraph. It is known that there is no $\omega(n^{-1/\operatorname{poly} \log \log n})$-approximation polynomial-time algorithm for the densest $k$-subgraph problem unless ETH fails (Manurangsi, 2017). We show how to transform an input of the densest $k$-subgraph problem to an input of FoN, and then show how to transform an approximate solution for FoN to an*

*approximate solution for the densest k-subgraph problem while only increasing the approximation factor by a constant. Therefore an $\omega(n^{-1/\text{poly}\log\log n})$-approximation polynomial-time algorithm for the FoN implies an $\omega(n^{-1/\text{poly}\log\log n})$-approximation polynomial-time algorithm for the densest k-subgraph problem, which does not exist unless ETH fails.*

To the best of our knowledge, this is the first subconstant hardness result for any RS problem. This is of particular importance because several previous works have chosen to optimize *surrogate functions*—these can be approximated well in theory—instead of the actual model accuracy. For instance, previous work considers a submodular surrogate function which can be approximated within a factor $1 - 1/e$ in polynomial time (Guillory, 2012; Wei et al., 2015; Mirzasoleiman et al., 2020; Chen & Krause, 2013; Esfandiari et al., 2021). Our hardness result implies the following.

**Corollary 1.** *In the worst case, there is a significant gap of $\omega(n^{-1/\text{poly}\log\log n})$ between $u(\mathcal{S}^*)$ and the solution of any polynomial-time approximable surrogate function that estimates $u(\mathcal{S}^*)$.*

The corollary follows because such surrogate functions can be optimized or approximated in polynomial time, but Theorem 1 shows that even for simple learning problems, approximating the accuracy is not possible in polynomial time (assuming the ETH); therefore, there are certain instances of the problem where optimizing for the surrogate functions does not optimize for the learning accuracy within the given approximation factor.

It is worth noting that many ML tasks are (known to be) hard to optimize, hence surrogate loss functions are commonly used in the context of deep learning: For example, finding a linear classifier with minimum 0-1 loss is NP-complete (Marcotte & Savard, 1992), and convex relaxations of this loss are typically used as surrogates in practice. What we show in this section is that fairly simple and natural instances of the representation selection problem are not only NP-complete but also hard to approximate. Thus the optimal solution of any surrogate function will be far from the actual optimum. This gap is prominent especially when we disentangle the task of finding the set to label from the task of training a model. We will see in the next sections that combining the two remedies the problem and produces very good results, which is in line with the applicability of surrogate loss functions within deep learning models.

We would like to clarify that the above hardness result holds for the general version of the problem. In fact, this hardness result does not apply when there are some structural assumption about the model. For example this does not apply when the learning model is linear, or when the number of features in the model is a constant. Therefore, this hardness result does not contradict with the previous approximation algorithms with such structural assumptions or limitations.

## 5 RSL in Presence of a Graph Structure

In the previous section, we established the hardness of RSL by finding a natural problem called FoN for which RSL is hard to approximate in polynomial time within any reasonable factor. The hardness of RSL motivates seeking other sources of information about the problem that may help better guide the selection process. We next study whether the presence of a graph structure in the case of GRSL can help tackle the hard RSL problem effectively, when the provided graph has a degree of homophily; that is, nodes belonging to the same class are more likely to be connected to each other than nodes belonging to different classes. We present an algorithm for GRSL and provide an empirical study. Note that our theoretical finding in Section 4 is a worst-case analysis and extends to the case of GRSL, because in the worst case the graph can be random and provide no extra information; connecting the hardness of GRSL to the degree of homophily of the provided graph is a much more difficult problem and we leave it as future work.

Let us start by providing a visual understanding of the latent space of the FoN problem. We construct a version of the FoN problem with $m = 1000$ data points $n = 10$ features, where we assume the first 5 features have type red and the next 5 features have type blue. For each data point, we select two of its features uniformly at random and set their values to a number from $[0.9, 1.0]$ (other values are 0). In Figure 2(a), we present a visualization of the data points using t-SNE (Van der Maaten & Hinton, 2008), where the colors represent the classes. From the visualization, we observe that there are 45 dense blocks of nodes (each having the same label) corresponding to $\binom{10}{2}$ different ways of selecting the position for the non-zero elements. The

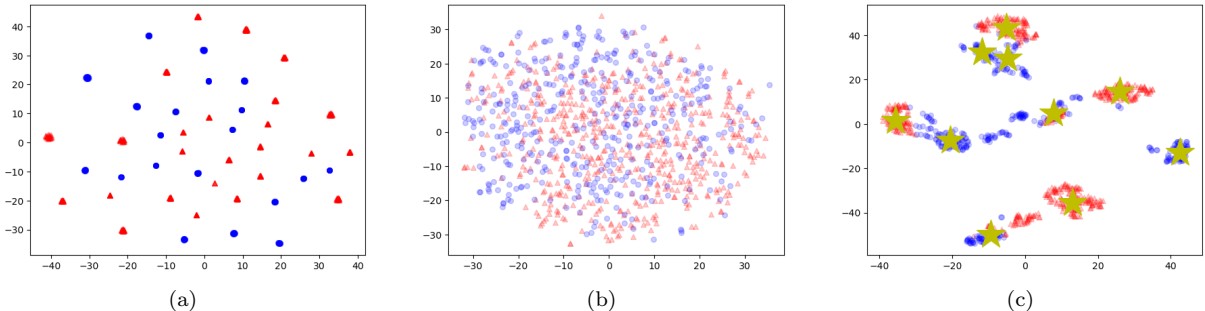

Figure 2: A t-SNE visualization of an instance of the FoN problem (the colors show the classes) when using (a) the original features, (b) adding a graph context and applying graph convolution, and (c) adding a graph context and applying a variant of RS-GNN to embed the nodes and select representatives (the yellow stars represent the selected representatives).

blocks are scattered in such a way that it is difficult for an RSL algorithm to select a small subset such that a model trained on the labels of the nodes in that subset generalizes well to the other data points. This analysis is similar to the use of the contextual Stochastic Block Model (cSBM) (Deshpande et al., 2018) which has been used for understanding GNNs through both theoretical analysis (Chien et al., 2021; Baranwal et al., 2021; 2023) and empirical study (Palowitch et al., 2022) in the supervised learning setting. However unlike a cSBM, we do not have a prior distribution assumption on the node features.

We now consider a variant of the FoN problem where a homophilous graph structure between the data points is also available.

**Definition 6** (GFoN). *GFoN is a variant of FoN where for each pair of data points we add an edge between them with probability $p$ if they belong to the same class and $p'$ if they belong to different classes. Letting $p > p'$ ensures the graph has some degree of homophily.*

Graph convolution is a popular modeling choice when a graph structure is available; it replaces the features for each node by a weighted average of the features of their neighbors, with weights proportional to the degrees. If we apply a graph convolution operation on GFoN with $p = 0.05$ and $p' = 0.01$, we get the updated node features in Figure 2(b). One can observe that the nodes from the two classes have a high overlap and so any RSL algorithm may still fail. To understand why this happens, consider four data points whose non-zero values are in positions 1 and 2, 9 and 10, 1 and 9, and 2 and 10 respectively. The first two nodes will have a label of 1 and the other two nodes will have a label of 0. However, if we disregard the node degrees, aggregating the first two nodes will give the same representation as aggregating the second two nodes.

The result in Figure 2(b) suggests that in presence of a graph structure, a naive application of graph convolution may not necessarily work best, and motivates the development of better modeling techniques. We show in our experiments that this also holds for many existing graph clustering/pooling approaches.

To better take advantage of the graph structure, in the next section we develop an approach that works by learning a mapping of the data points to a latent space where 1- we can group the nodes and select representatives that cover groups of nodes, and 2- we can better distinguish nodes belonging to each class. Figure 2(c) shows a t-SNE visualization of the data points in the latent space and the selected representatives. One can observe that with appropriate modeling, the presence of a homophilous graph structure turns the provably hard FoN problem into an RSL problem that can be effectively solved.

We next describe our approach for leveraging the graph context for the GRSL problem. Our theoretical and empirical results motivate obtaining or constructing graph contexts for RSL problems when possible.

# 6 RS-GNN: A Representation Learning-based Model for GRSL

We observed that while a (homophilous) graph structure can be quite helpful for RSL and make hard problems effectively solvable, for this to happen one needs an appropriate modeling approach. To this end, in this section we develop **RS-GNN**, a representation learning-based approach for **RS** via **GNN**s.

**Graph Neural Networks (GNNs)** encode graph-structured data in continuous space (Chami et al., 2020). Graph convolutional networks (GCNs) (Kipf & Welling, 2016) are a powerful variant of GNNs. Let $\mathcal{G} = \{\mathcal{V}, \boldsymbol{A}, \boldsymbol{X}\}$ be an attributed graph, $\hat{\boldsymbol{A}} = \boldsymbol{A} + \boldsymbol{I}$ be the adjacency matrix of $\mathcal{G}$ with self-loops included, and $\boldsymbol{D}$ be the degree matrix of $\hat{\boldsymbol{A}}$, where $\boldsymbol{D}_{ii} = \sum_j \hat{\boldsymbol{A}}_{ij}$ and $\boldsymbol{D}_{ij} = 0$ for $i \neq j$. The $l^{\text{th}}$ layer of an L-layer GCN model with parameters $\boldsymbol{\Theta} = \{\boldsymbol{W}^{(1)}, \dots, \boldsymbol{W}^{(L)}\}$ can be defined as: $\boldsymbol{H}^{(l)} = \sigma(\boldsymbol{D}^{-\frac{1}{2}} \hat{\boldsymbol{A}} \boldsymbol{D}^{-\frac{1}{2}} \boldsymbol{H}^{(l-1)} \boldsymbol{W}^{(l)})$, where $\boldsymbol{H}^{(l)}$ represents node embeddings in the $l^{\text{th}}$ layer ($\boldsymbol{H}^{(0)} = \boldsymbol{X}$), $\boldsymbol{W}^{(l)}$ is a weight matrix, and $\sigma$ is an activation function. In the rest of the paper, we use $\mathsf{GCN}(\mathcal{G}; \boldsymbol{\Theta})$ to show the application of a GCN function with parameters $\boldsymbol{\Theta}$ on a graph $\mathcal{G}$.

**Deep Graph Infomax (DGI)** (Velickovic et al., 2019) is an approach for unsupervised representation-learning in attributed graphs. Given an attributed graph $\mathcal{G}$, in each iteration DGI creates a corrupted graph $\mathcal{G}'$ from $\mathcal{G}$. Then, it computes node embeddings $\boldsymbol{H}$ and $\boldsymbol{H}'$ for the two graphs by applying a GNN model on them, and a summary vector $\boldsymbol{s}$ based on the node embeddings $\boldsymbol{H}$. Finally, a discriminator is simultaneously trained to separate the node embeddings of the original graph (i.e., $\boldsymbol{H}$) from those of the corrupted graph (i.e., $\boldsymbol{H}'$) based on the summary vector $\boldsymbol{s}$. We describe the details of each step later when we define our final model.

## 6.1 Representative Selection via GNNs

An attributed graph $\mathcal{G} = \{\mathcal{V}, \boldsymbol{A}, \boldsymbol{X}\}$ has two modalities, the node features $\boldsymbol{X}$ and the graph structure $\boldsymbol{A}$. To be able to exploit both modalities in selecting good representatives, we employ a function $\mathsf{EMB}$ that combines the two modalities into a single embedding matrix of the graph. Concretely, $\mathsf{EMB}(\mathcal{G}) = \boldsymbol{H}$, where $\boldsymbol{H} \in \mathbb{R}^{m \times d}$ and $d$ represents the embedding dimension. We also employ a differentiable function $\mathsf{SEL}$ that receives $\boldsymbol{H}$ as input and selects $k$ nodes as representatives. That is, $\mathsf{SEL}(\boldsymbol{H}) = \mathcal{S}$. The two functions are optimized in a multi-task setting with the loss function

$$\mathcal{L} = \mathcal{L}_{\mathsf{EMB}} + \lambda \mathcal{L}_{\mathsf{SEL}}, \tag{1}$$

where $\mathcal{L}_{\mathsf{EMB}}$ encourages learning informative node embeddings and $\mathcal{L}_{\mathsf{SEL}}$ encourages selecting good representatives. One can create different RSL models with different choices of $\mathsf{EMB}$, $\mathsf{SEL}$, $\mathcal{L}_{\mathsf{EMB}}$ and $\mathcal{L}_{\mathsf{SEL}}$. GNNs have proved effective in learning node embeddings, so we use GNNs as our embedding function $\mathsf{EMB}$. For $\mathcal{L}_{\mathsf{EMB}}$, we use the DGI objective which has shown to provide high-quality embeddings in unsupervised settings. For $\mathsf{SEL}$, we consider a representative embedding matrix $\boldsymbol{R} \in \mathbb{R}^{k \times d}$ with learnable parameters where $\boldsymbol{R}$ is initialized randomly and $\boldsymbol{R}_j$ represents the embedding for the $j^{\text{th}}$ representative. We let

$$\mathcal{L}_{\mathsf{SEL}} = \sum_i \min_j(\mathsf{Dist}(\boldsymbol{H}_i, \boldsymbol{R}_j)), \tag{2}$$

where $\mathsf{Dist}(\boldsymbol{H}_i, \boldsymbol{R}_j)$ is the distance between the $i^{\text{th}}$ node's embedding $\boldsymbol{H}_i$ and the $j^{\text{th}}$ representative's embedding $\boldsymbol{R}_j$. We use Euclidean distance as the distance function. We select the representative corresponding to each $\boldsymbol{R}_j$ by finding the closest node embedding from $\boldsymbol{H}$ to $\boldsymbol{R}_j$, i.e., $\mathsf{argmin}_i(\mathsf{Dist}(\boldsymbol{H}_i, \boldsymbol{R}_j))$.

Observe that with the loss function of equation 1, the model can trivially reduce $\mathcal{L}_{\mathsf{SEL}}$ by making the values in $\boldsymbol{H}$ arbitrarily small. That is because multiplying $\boldsymbol{H}$ by a small constant may not change $\mathcal{L}_{\mathsf{EMB}}$ substantially, but it can make the distances between the nodes arbitrarily small, resulting in a low value for $\mathcal{L}_{\mathsf{SEL}}$ even for a random representative embedding matrix $\boldsymbol{R}$. In the next subsection we describe a normalization scheme, *CenterNorm*, that is applied to $\boldsymbol{H}$ before $\boldsymbol{H}$ is used in equation 2, which helps avoid this problem.

## 6.2 CenterNorm

As we show in Appendix B, using the DGI loss function results in corrupted node embeddings that form a dense cluster in some part of the embedding space, and node embeddings (from the actual graph) that arrange themselves in subclusters around (and outside) this dense cluster of negative examples. Based on the this observation, we propose an $\ell_2$ normalization of the node embeddings in $\boldsymbol{H}$ with respect to the center of the node embeddings:

$$\boldsymbol{\mu} = \frac{1}{n}\sum_i \boldsymbol{H}_i,\ \boldsymbol{\zeta} = \|\boldsymbol{H} - \boldsymbol{\mu}\|,\ \tilde{\boldsymbol{H}} = (\boldsymbol{H} - \boldsymbol{\mu})/\boldsymbol{\zeta}, \tag{3}$$

where $\boldsymbol{\mu}$ is the center of the embeddings $\boldsymbol{H}$, $\boldsymbol{\zeta}$ is the $\ell_2$ norms of the nodes with respect to the center, and $\tilde{\boldsymbol{H}}$ represents the normalized embeddings. With CenterNorm, the model can no longer decrease $\mathcal{L}_{\mathsf{SEL}}$ simply by making the values $\boldsymbol{H}$ smaller. Note that since the embedding clusters in $\boldsymbol{H}$ are at a large angle from each other (cf. supplementary material), $\ell_2$ normalization has a low chance of collapsing two clusters. Furthermore, $\ell_2$ normalization helps bring the nodes within one cluster closer to each other, which helps in identifying clusters and selecting representatives.

## 6.3 The Final Model: RS-GNN

The full RS-GNN model is described in Algorithm 1. The input is an attributed graph $\mathcal{G} = (\mathcal{V}, \boldsymbol{A}, \boldsymbol{X})$ and a number $k$ corresponding to the number of representative nodes that must be selected from the graph. The model initializes $\boldsymbol{R}$ (the representative embeddings), $\boldsymbol{\Theta}$ (the GCN parameters), and $\boldsymbol{U}$ (the parameters for the DGI discriminator). Lines 3 to 7 compute node embeddings $\boldsymbol{H}$ using a GCN [3] model and compute a DGI loss as $\mathcal{L}_{\mathsf{EMB}}$. Here, $\mathsf{bilinear}(\boldsymbol{H}, \boldsymbol{s}; \boldsymbol{U}) = \mathsf{sigmoid}(\boldsymbol{H}^T \boldsymbol{U} \boldsymbol{s})$ where $\boldsymbol{H}^T$ indicates the transpose of $\boldsymbol{H}$. Lines 8 and 9 apply CenterNorm. Lines 10 to 12 compute $\mathcal{L}_{\mathsf{SEL}}$ based on equation 2 and then $\mathcal{L}$ based on equation 1 and update the parameters accordingly.

We keep track of the epoch with minimum joint loss $\mathcal{L}$. Let $\hat{\boldsymbol{R}}$ and $\hat{\boldsymbol{H}}$ be the corresponding representative and normalized node embeddings in the best epoch. We select the $j^{\text{th}}$

---

**Algorithm 1** The training procedure of RS-GNN.

**Input:** $\mathcal{G} = (\mathcal{V}, \boldsymbol{A}, \boldsymbol{X})$, $k$
1: Initialize $\boldsymbol{R}$, $\boldsymbol{\Theta}$, and $\boldsymbol{U}$
2: **for** epoch=1 **to** #epochs **do**
3: $\quad \mathcal{G}' = (\mathcal{V}, \boldsymbol{A}, \mathsf{shuffle}(\boldsymbol{X}))$
4: $\quad \boldsymbol{H} = \mathsf{GCN}(\mathcal{G}; \boldsymbol{\Theta}), \quad \boldsymbol{H}' = \mathsf{GCN}(\mathcal{G}'; \boldsymbol{\Theta})$
5: $\quad \boldsymbol{s} = \mathsf{sigmoid}(\frac{1}{n}\sum_i \boldsymbol{H}_i)$
6: $\quad \boldsymbol{p} = \mathsf{bilinear}(\boldsymbol{H}, \boldsymbol{s}; \boldsymbol{U}), \ \boldsymbol{p}' = \mathsf{bilinear}(\boldsymbol{H}', \boldsymbol{s}; \boldsymbol{U})$
7: $\quad \mathcal{L}_{\mathsf{EMB}} = -\sum_i (log(\boldsymbol{p}_i) + log(1 - \boldsymbol{p}'_i))$
8: $\quad \boldsymbol{\mu} = \frac{1}{n}\sum_i \boldsymbol{H}_i, \boldsymbol{\zeta} = \|\boldsymbol{H} - \boldsymbol{\mu}\|$
9: $\quad \tilde{\boldsymbol{H}} = \mathsf{CenterNorm}(\boldsymbol{H}) = (\boldsymbol{H} - \boldsymbol{\mu})/\boldsymbol{\zeta}$
10: $\quad \mathcal{L}_{\mathsf{SEL}} = \sum_i \min_j \mathsf{Dist}(\tilde{\boldsymbol{H}}_i, \boldsymbol{R}_j)$
11: $\quad \mathcal{L} = \mathcal{L}_{\mathsf{EMB}} + \lambda \mathcal{L}_{\mathsf{SEL}}$
12: $\quad$ Compute gradients for $\mathcal{L}$, upd. params.
13: Let $\hat{\boldsymbol{R}}$ and $\hat{\boldsymbol{H}}$ be the representative and normalized node embeddings with minimum $\mathcal{L}$ during training.
14: **for** j=1 **to** k **do**
15: $\quad$ The $j^{\text{th}}$ representative = $\mathsf{argmin}_i \mathsf{Dist}(\hat{\boldsymbol{H}}_i, \hat{\boldsymbol{R}}_j)$

---

representative to be the node whose normalized embedding is closest to the $j^{\text{th}}$ representative embedding. Lines 13 to 15 select the representatives based on $\hat{\boldsymbol{R}}$ and $\hat{\boldsymbol{H}}$.

**Generality:** By pairing unsupervised learning approaches with appropriate normalization schemes (such as DGI and CenterNorm in this paper), one can extend the formulation in equation 1 to other unsupervised graph representation learning approaches and even to other data types (e.g., images or text). We leave this as future work.

## 6.4 RS-GNN for GFoN

In Figure 2(c), we showed that RS-GNN can effectively solve the GFoN. To understand why this happens, notice that the discriminator in Algorithm 1 needs to distinguish the node embeddings $\boldsymbol{H}$ from $\boldsymbol{H}'$. Since we

---

[3] While we use GCN for direct comparability to existing work, note that one can use any other GNN model for RS-GNN. For example, for the FoN problem in Figure 2(c), we used a variant of the GraphSage (Hamilton et al., 2017) model as it better matched the problem.

constructed a homophilous graph structure, each embedding in $\boldsymbol{H}$ is the aggregation of a set of nodes mostly belonging to one class, whereas each embedding in $\boldsymbol{H'}$ is the aggregation of a set of nodes that may belong to any of the classes. In order for the discriminator to be able to distinguish these two cases, the GNN model needs to learn to map the data to a space where nodes belonging to each class are separate from each other; otherwise the aggregation of nodes (mostly) from one class may be similar to the aggregation of nodes from both classes. Moreover, the addition of $\mathcal{L}_{\mathsf{SEL}}$ motivates the model to group the nodes around a few center points giving rise to the dense groups of same-class nodes in Figure 2.

## 7 Empirical Results

We describe our baselines, datasets, and metrics, and defer implementation details to supplementary material.[4]

**Baselines:** We compare against a representative set of baselines from different categories, as described below. The details for each baseline can be found in Appendix D.

- *Random:* Selects $k$ nodes uniformly at random.

- *Popular:* Selects the $k$ nodes with the maximum degree from the graph.

- *Surrogate functions on node features:* We test a representative set of surrogate functions: KMedoid, KMeans, Farthest First Search (FFS) (You et al., 2020) and (Greedy) MaxCover (Hochbaum & Pathria, 1998). For KMeans, we select the closest node to each cluster center as a representative. For FFS and MaxCover, we select representatives sequentially. In FFS , the next representative is the node farthest away (by Euclidean distance) from the closest representative in the current set. In MaxCover , the next representative is the node that increases the coverage of the non-selected nodes the most. For MaxCover, we experiment with RBF kernel and cosine similarities; we use MC-RBF and MC-Cos to refer to the two versions respectively. Note that the sequential nature of FFS and MC makes them less amenable to parallelization.

- *Surrogate functions on node embeddings:* We use similar functions as above but apply them on DGI node embeddings as opposed to on the initial node features. Note that when we run these baselines using DGI embeddings as context, their selections are informed by both node features and the graph structure. Following the Spectral Clustering (Ng et al., 2001) paradigm, we also experiment with KMeans on spectral embeddings (i.e. top singular vectors) in two settings: 1- based on the graph Laplacian, and 2- based on the kNN affinity graph of the node similarities.

- *Graph clustering/pooling/active learning:* We compare against a number of graph clustering/pooling/active learning approaches from different categories. Specifically, we compare against MinCut (Bianchi et al., 2020) which is a well-established pooling approach, FeatProp (Wu et al., 2019b) which is successful graph active learning approaches, SDCN (Bo et al., 2020) which is a well-established auto-encoder based graph clustering model, EGAE (Fettal et al., 2022) and GCC (Fettal et al., 2022) which are recent joint representation learning and clustering approaches, and DMoN (Tsitsulin et al., 2023) which is a state-of-the-art graph clustering approach based on modularity maximization.

**Datasets:** We use eight established benchmarks in the GNN literature: three citation networks namely Cora, CiteSeer, and Pubmed Sen et al. (2008); Hu et al. (2020), a citation network named OGBN-Arxiv Hu et al. (2020) which is orders of magnitude larger than the previous three, two datasets from Amazon products (Photos and PC) Shchur et al. (2018), and two datasets from Microsoft Academic (CS and physics) Shchur et al. (2018). Supplementary material offers a more detailed description of datasets and their statistics. Our datasets have a wide range in terms of the number of nodes (from 2K to 170K), edges (from 4.5K to 1.1M), features (from 100 to 8.5K) and classes (from 3 to 40).

**Measures:** We measure the quality of the selected representatives $\mathcal{S} \subseteq \mathcal{V}$ using the following transductive semi-supervised node-classification problem. We train a GCN model on the dataset where the parameters of the GCN are learned only based on the labels of the nodes in $\mathcal{S}$. Note that this GCN is completely

---

[4]The code is available at: `https://github.com/google-research/google-research/tree/master/rs_gnn`.

Table 1: For each dataset, each algorithm selects $2c$ representatives. Then, we train a GCN model on the labels of the selected representatives. The reported metric is the test accuracy of the GCN models. Bold numbers indicate statistically significant winner(s) following a t-test (p-value=0.05). The color-codes and symbols represent ⁜ surrogate functions on features, ♣ surrogate functions on embeddings, and ❦ attributed graph clustering/pooling or active learning approaches.

| | Selector | Context | Cora | CiteSeer | Pubmed | Photos |
|---|---|---|---|---|---|---|
| | Random | — | $49.1_{\pm 6.9}$ | $33.1_{\pm 8.3}$ | $52.0_{\pm 8.1}$ | $70.2_{\pm 6.4}$ |
| | Popular | **A** | $59.2_{\pm 1.3}$ | $35.5_{\pm 0.8}$ | $63.3_{\pm 0.3}$ | $34.9_{\pm 0.5}$ |
| ⁜ | KMedoid | **X** | $53.7_{\pm 5.4}$ | $40.3_{\pm 2.8}$ | $53.2_{\pm 1.0}$ | $65.8_{\pm 1.2}$ |
| ⁜ | KMeans | **X** | $32.5_{\pm 5.8}$ | $35.4_{\pm 1.2}$ | $50.6_{\pm 0.5}$ | $72.5_{\pm 0.9}$ |
| ⁜ | FFS | **X** | $48.5_{\pm 8.0}$ | $39.3_{\pm 6.8}$ | $43.1_{\pm 4.9}$ | $80.0_{\pm 5.0}$ |
| ⁜ | MC-RBF | **X** | $45.5_{\pm 2.7}$ | $25.8_{\pm 3.7}$ | $53.0_{\pm 0.2}$ | $78.4_{\pm 1.2}$ |
| ⁜ | MC-Cos | **X** | $49.7_{\pm 9.5}$ | $50.2_{\pm 3.1}$ | $66.6_{\pm 0.5}$ | $77.2_{\pm 1.0}$ |
| ♣ | KMedoid | **DGI** | $48.4_{\pm 4.4}$ | $34.1_{\pm 1.9}$ | $60.9_{\pm 5.3}$ | $81.5_{\pm 2.6}$ |
| ♣ | KMeans | **DGI** | $62.6_{\pm 9.3}$ | $42.7_{\pm 6.3}$ | $60.5_{\pm 6.3}$ | $83.6_{\pm 2.9}$ |
| ♣ | KMeans | **Spectral(X)** | $48.8_{\pm 0.4}$ | $19.9_{\pm 2.4}$ | $45.1_{\pm 0.4}$ | $81.8_{\pm 0.5}$ |
| ♣ | KMeans | **Spectral(A)** | $56.8_{\pm 0.4}$ | $30.1_{\pm 5.2}$ | $57.1_{\pm 1.0}$ | $46.9_{\pm 0.6}$ |
| ♣ | FFS | **DGI** | $62.6_{\pm 4.5}$ | $50.4_{\pm 5.7}$ | $46.7_{\pm 7.2}$ | $73.4_{\pm 5.7}$ |
| ♣ | MC-RBF | **DGI** | $66.3_{\pm 2.6}$ | $35.3_{\pm 4.5}$ | $54.9_{\pm 5.3}$ | $37.4_{\pm 3.7}$ |
| ♣ | MC-Cos | **DGI** | $67.3_{\pm 5.2}$ | $49.0_{\pm 4.1}$ | $\mathbf{67.3}_{\pm 1.1}$ | $84.4_{\pm 1.0}$ |
| ❦ | MinCUT | **X, A** | $51.9_{\pm 7.5}$ | $37.3_{\pm 8.0}$ | $59.5_{\pm 6.1}$ | $14.4_{\pm 7.5}$ |
| ❦ | FeatProp | **X, A** | $56.6_{\pm 1.7}$ | $37.8_{\pm 1.6}$ | $65.2_{\pm 0.6}$ | $78.2_{\pm 1.8}$ |
| ❦ | SDCN | **X, A** | $41.6_{\pm 9.5}$ | $33.8_{\pm 9.3}$ | $47.8_{\pm 8.5}$ | $61.0_{\pm 10.8}$ |
| ❦ | EGAE | **X, A** | $64.4_{\pm 3.8}$ | $45.0_{\pm 5.8}$ | $57.7_{\pm 5.1}$ | $83.5_{\pm 3.0}$ |
| ❦ | GCC | **X, A** | $68.7_{\pm 2.2}$ | $49.3_{\pm 6.0}$ | $63.9_{\pm 5.1}$ | $84.2_{\pm 1.4}$ |
| ❦ | DMoN | **X, A** | $58.0_{\pm 7.1}$ | $40.5_{\pm 7.4}$ | $55.3_{\pm 7.5}$ | $78.6_{\pm 9.2}$ |
| | RS-GNN | **X, A** | $\mathbf{72.4}_{\pm 3.7}$ | $\mathbf{54.7}_{\pm 3.9}$ | $65.8_{\pm 3.0}$ | $\mathbf{86.3}_{\pm 1.4}$ |

independent of the internal GCN model used in RS-GNN. We randomly split the remaining nodes into validation and test sets. The validation set is used for early stopping. The classification accuracy on the test set is used as the metric for measuring the quality of the selected representatives. Considering a validation set for early stopping reduces the chances of overfitting for the classifier and makes the reported test accuracy mainly a function of the quality of the selected representatives.

### 7.1 GRSL Empirical Results with Natural Graphs

The results are presented in Table 1. For the results in this table, we set $k$ for each dataset to be $2c$, where $c$ represents the number of classes. We found this to be a small enough number for a meaningful comparison of the quality of the selected representatives [5], and high enough for the classification GCN model to learn appropriate functions of the data. Since $c$ is different for each dataset, making $k$ a function of $c$ also provides the opportunity to compare performance not only in terms of variation in the datasets, but also in terms of variation in the number of selected representatives.

RS-GNN performs well across all datasets and consistently outperforms (or matches) the baselines (with the exception of Pubmed). It has a low variance across different runs making it a reliable model. Among the surrogate functions, MC-Cos and KMeans perform best. We found FFS to be sensitive to outliers. We also found it difficult to select a set of hyperparameters for MC-RBF that work well across datasets. Among graph clustering/pooling and active learning approaches, we found GCC to perform best and be the only model that (overall) outperforms surrogate functions when applied on DGI embeddings. Many of the other graph clustering/pooling or active learning approaches even fall short of the MC-Cos model when applied on the node features alone, thus showing the importance of developing appropriate models for taking advantage

---

[5]Note that if $k \approx n$, all models may perform equally well.

Table 2: For each dataset, each algorithm selects 2*c* representatives. Then, we train a GCN model on the labels of the selected representatives. The reported metric is the test accuracy of the GCN models. Bold numbers indicate statistically significant winner(s) following a t-test (p-value=0.05). OOM indicates out of memory.

| | Selector | Context | PC | CS | Physics | Arxiv | Avg. |
|---|---|---|---|---|---|---|---|
| | Random | — | $65.4_{\pm6.1}$ | $72.9_{\pm5.0}$ | $73.6_{\pm6.8}$ | $49.0_{\pm1.4}$ | 58.2 |
| | Popular | **A** | $49.9_{\pm2.1}$ | $73.1_{\pm1.1}$ | $50.5_{\pm0.0}$ | $31.3_{\pm0.8}$ | 49.7 |
| ✤ | KMedoid | **X** | $66.9_{\pm1.8}$ | $51.8_{\pm1.0}$ | $66.6_{\pm1.6}$ | $43.9_{\pm1.5}$ | 55.3 |
| ✤ | KMeans | **X** | $70.9_{\pm1.0}$ | $66.5_{\pm0.5}$ | $73.0_{\pm1.4}$ | $48.4_{\pm0.5}$ | 56.2 |
| ✤ | FFS | **X** | $71.5_{\pm3.4}$ | $54.6_{\pm2.2}$ | $76.6_{\pm2.8}$ | $45.2_{\pm0.8}$ | 57.3 |
| ✤ | MC-RBF | **X** | $65.5_{\pm0.9}$ | $66.4_{\pm1.2}$ | $58.1_{\pm0.3}$ | $51.4_{\pm0.6}$ | 55.5 |
| ✤ | MC-Cos | **X** | $72.4_{\pm0.8}$ | $79.3_{\pm0.8}$ | $88.3_{\pm1.0}$ | $50.0_{\pm0.5}$ | 66.7 |
| ✤ | KMedoid | **DGI** | $69.8_{\pm3.4}$ | $82.5_{\pm2.9}$ | $81.2_{\pm7.0}$ | OOM | — |
| ✤ | KMeans | **DGI** | $\mathbf{74.8_{\pm2.7}}$ | $86.9_{\pm1.8}$ | $\mathbf{90.6_{\pm2.4}}$ | $51.2_{\pm1.0}$ | 69.1 |
| ✤ | KMeans | **Spectral(X)** | $68.3_{\pm2.3}$ | $68.5_{\pm1.1}$ | $87.7_{\pm0.2}$ | OOM | — |
| ✤ | KMeans | **Spectral(A)** | $54.1_{\pm1.5}$ | $76.8_{\pm0.3}$ | $50.5_{\pm0.0}$ | OOM | — |
| ✤ | FFS | **DGI** | $63.5_{\pm6.4}$ | $84.8_{\pm5.3}$ | $83.4_{\pm5.0}$ | $48.6_{\pm2.2}$ | 64.2 |
| ✤ | MC-RBF | **DGI** | $50.7_{\pm2.5}$ | $65.2_{\pm1.2}$ | $59.8_{\pm5.1}$ | $41.2_{\pm1.6}$ | 51.4 |
| ✤ | MC-Cos | **DGI** | $74.0_{\pm3.7}$ | $87.3_{\pm1.4}$ | $81.3_{\pm3.4}$ | $47.6_{\pm1.6}$ | 70.0 |
| ❦ | MinCUT | **X, A** | $18.3_{\pm8.7}$ | $85.5_{\pm1.4}$ | $86.0_{\pm3.3}$ | $32.4_{\pm5.2}$ | 48.2 |
| ❦ | FeatProp | **X, A** | $68.5_{\pm1.1}$ | $74.7_{\pm0.3}$ | $81.4_{\pm0.6}$ | $47.8_{\pm1.0}$ | 63.8 |
| ❦ | SDCN | **X, A** | $54.2_{\pm8.7}$ | $66.4_{\pm6.7}$ | $77.5_{\pm9.2}$ | $38.4_{\pm4.4}$ | 52.6 |
| ❦ | EGAE | **X, A** | $\mathbf{75.4_{\pm3.2}}$ | $79.9_{\pm3.2}$ | $80.4_{\pm4.0}$ | $50.6_{\pm1.0}$ | 67.1 |
| ❦ | GCC | **X, A** | $72.1_{\pm2.1}$ | $85.8_{\pm1.5}$ | $\mathbf{89.6_{\pm0.9}}$ | $\mathbf{52.1_{\pm1.4}}$ | 70.7 |
| ❦ | DMoN | **X, A** | $70.3_{\pm3.3}$ | $84.3_{\pm1.4}$ | $85.9_{\pm3.8}$ | $\mathbf{52.5_{\pm1.9}}$ | 65.7 |
| | RS-GNN | **X, A** | $\mathbf{74.3_{\pm1.7}}$ | $\mathbf{89.3_{\pm0.8}}$ | $\mathbf{90.0_{\pm2.6}}$ | $\mathbf{52.6_{\pm1.2}}$ | **73.2** |

Table 3: Classification accuracies when a graph structure is not provided as input. Selecting 5*c* representatives. Bold numbers indicate statistically significant winner(s) following a t-test (p-value=0.05).

| Selector | Model | Cora | Citeseer | Pubmed | Photos | PC | CS | Physics | Avg. |
|---|---|---|---|---|---|---|---|---|---|
| Random | MLP | $41.9_{\pm2.9}$ | $37.4_{\pm4.0}$ | $51.9_{\pm5.0}$ | $57.5_{\pm4.0}$ | $55.2_{\pm4.4}$ | $76.1_{\pm2.7}$ | $76.7_{\pm5.2}$ | 56.7 |
| Random | GCN | $57.7_{\pm4.1}$ | $57.7_{\pm4.5}$ | $59.6_{\pm4.6}$ | $79.2_{\pm3.4}$ | $71.8_{\pm4.4}$ | $84.8_{\pm2.2}$ | $86.0_{\pm2.9}$ | 71.0 |
| KMedoid | MLP | $39.3_{\pm0.9}$ | $33.4_{\pm0.7}$ | $46.1_{\pm1.0}$ | $40.6_{\pm0.9}$ | $47.5_{\pm0.8}$ | $62.8_{\pm1.6}$ | $67.0_{\pm0.8}$ | 48.1 |
| KMedoid | GCN | $52.5_{\pm1.4}$ | $57.4_{\pm0.8}$ | $55.0_{\pm0.7}$ | $72.3_{\pm0.6}$ | $63.4_{\pm1.8}$ | $66.2_{\pm2.9}$ | $70.8_{\pm0.3}$ | 62.5 |
| KMeans | MLP | $42.4_{\pm0.9}$ | $40.1_{\pm1.1}$ | $58.5_{\pm1.3}$ | $70.0_{\pm1.0}$ | $63.7_{\pm0.8}$ | $68.5_{\pm0.7}$ | $77.4_{\pm0.3}$ | 60.1 |
| KMeans | GCN | $56.5_{\pm0.8}$ | $55.9_{\pm0.8}$ | $\mathbf{72.7_{\pm0.3}}$ | $80.9_{\pm0.6}$ | $\mathbf{75.8_{\pm0.6}}$ | $80.0_{\pm0.4}$ | $83.7_{\pm0.8}$ | 72.2 |
| FFS | MLP | $39.2_{\pm3.7}$ | $44.0_{\pm3.1}$ | $43.1_{\pm3.0}$ | $60.2_{\pm3.8}$ | $55.8_{\pm1.9}$ | $56.3_{\pm0.9}$ | $77.7_{\pm2.5}$ | 53.8 |
| FFS | GCN | $56.9_{\pm2.6}$ | $62.3_{\pm3.7}$ | $48.8_{\pm3.9}$ | $80.7_{\pm2.0}$ | $\mathbf{74.8_{\pm2.5}}$ | $59.8_{\pm2.5}$ | $84.0_{\pm2.1}$ | 66.8 |
| MC-Cos | MLP | $46.5_{\pm2.2}$ | $47.5_{\pm2.3}$ | $52.1_{\pm0.7}$ | $54.0_{\pm2.4}$ | $58.2_{\pm1.2}$ | $83.0_{\pm0.5}$ | $86.8_{\pm0.7}$ | 61.2 |
| MC-Cos | GCN | $62.0_{\pm1.7}$ | $\mathbf{63.0_{\pm2.5}}$ | $59.3_{\pm1.3}$ | $79.5_{\pm0.5}$ | $\mathbf{75.4_{\pm1.1}}$ | $\mathbf{87.6_{\pm0.3}}$ | $\mathbf{92.8_{\pm0.3}}$ | 74.2 |
| RS-GNN | GCN | $\mathbf{64.6_{\pm2.4}}$ | $\mathbf{64.3_{\pm2.1}}$ | $65.1_{\pm3.0}$ | $\mathbf{82.2_{\pm2.0}}$ | $75.5_{\pm2.1}$ | $88.3_{\pm1.6}$ | $89.9_{\pm1.9}$ | **75.7** |

of the graph structure and confirming our finding in Figure 2. MinCut produced degenerate solutions for Photos and PC datasets in many runs (assigning all nodes to one cluster), hence performing poorly on them.

## 7.2 GRSL Empirical Results with Similarity Graphs

In several applications, a natural graph may not be available. For semi-supervised node classification, it has recently been shown that even without a natural graph, one can still leverage GNNs by learning both a graph structure and GNN parameters simultaneously (Fatemi et al., 2021; 2023). We extend the aforementioned

Table 4: Label coverage of different RS algorithms when $k = 2c$. The surrogate function selectors marked in blue use DGI embeddings.

| Selector | Cora | CiteSeer | Pubmed | Photos | PC | CS | Physics | Arxiv |
|---|---|---|---|---|---|---|---|---|
| Random | $86.4_{\pm10.8}$ | $84.2_{\pm12.7}$ | $85.0_{\pm17.0}$ | $80.6_{\pm11.1}$ | $64.0_{\pm9.4}$ | $72.7_{\pm9.4}$ | $75.0_{\pm14.3}$ | $55.0_{\pm4.9}$ |
| Popular | $85.7_{\pm0.0}$ | $50.0_{\pm0.0}$ | $66.7_{\pm0.0}$ | $37.5_{\pm0.0}$ | $30.0_{\pm0.0}$ | $66.7_{\pm0.0}$ | $20.0_{\pm0.0}$ | $15.0_{\pm0.0}$ |
| KMedoid | $100.0_{\pm0.0}$ | $100.0_{\pm0.0}$ | $66.7_{\pm0.0}$ | $87.5_{\pm0.0}$ | $80.0_{\pm0.0}$ | $73.3_{\pm0.0}$ | $60.0_{\pm0.0}$ | $50.0_{\pm0.0}$ |
| KMeans | $100.0_{\pm0.0}$ | $83.3_{\pm0.0}$ | $100.0_{\pm0.0}$ | $75.0_{\pm0.0}$ | $70.0_{\pm0.0}$ | $46.7_{\pm0.0}$ | $60.0_{\pm0.0}$ | $65.0_{\pm0.0}$ |
| FFS | $84.3_{\pm11.3}$ | $88.3_{\pm9.5}$ | $71.7_{\pm12.2}$ | $93.1_{\pm8.6}$ | $74.5_{\pm9.4}$ | $36.3_{\pm3.4}$ | $67.0_{\pm9.8}$ | $63.0_{\pm2.3}$ |
| MC-RBF | $100.0_{\pm0.0}$ | $100.0_{\pm0.0}$ | $66.7_{\pm0.0}$ | $75.0_{\pm0.0}$ | $60.0_{\pm0.0}$ | $66.7_{\pm0.0}$ | $60.0_{\pm0.0}$ | $52.5_{\pm0.0}$ |
| MC-Cos | $97.1_{\pm5.9}$ | $98.3_{\pm5.1}$ | $100.0_{\pm0.0}$ | $75.0_{\pm0.0}$ | $80.0_{\pm0.0}$ | $66.7_{\pm0.0}$ | $100.0_{\pm0.0}$ | $55.2_{\pm0.8}$ |
| KMedoid | $79.3_{\pm11.8}$ | $58.3_{\pm10.1}$ | $93.3_{\pm13.7}$ | $100.0_{\pm0.0}$ | $80.0_{\pm9.2}$ | $83.7_{\pm7.0}$ | $87.0_{\pm14.9}$ | OOM |
| KMeans | $100.0_{\pm0.0}$ | $90.0_{\pm8.4}$ | $100.0_{\pm0.0}$ | $96.9_{\pm5.5}$ | $82.5_{\pm6.4}$ | $99.3_{\pm2.0}$ | $100.0_{\pm0.0}$ | $54.2_{\pm5.0}$ |
| FFS | $96.4_{\pm6.3}$ | $86.7_{\pm10.3}$ | $66.7_{\pm18.7}$ | $77.5_{\pm7.7}$ | $65.0_{\pm6.1}$ | $97.0_{\pm3.4}$ | $95.0_{\pm8.9}$ | $52.0_{\pm5.1}$ |
| MC-RBF | $71.4_{\pm0.0}$ | $81.7_{\pm5.1}$ | $66.7_{\pm0.0}$ | $75.0_{\pm0.0}$ | $60.0_{\pm0.0}$ | $52.0_{\pm4.6}$ | $85.0_{\pm8.9}$ | $28.1_{\pm4.0}$ |
| MC-Cos | $100.0_{\pm0.0}$ | $100.0_{\pm0.0}$ | $100.0_{\pm0.0}$ | $87.5_{\pm0.0}$ | $82.5_{\pm5.5}$ | $93.7_{\pm1.5}$ | $85.0_{\pm11.0}$ | $37.5_{\pm2.4}$ |
| MinCUT | $78.6_{\pm8.7}$ | $83.3_{\pm15.3}$ | $95.0_{\pm12.2}$ | $12.5_{\pm0.0}$ | $10.0_{\pm0.0}$ | $85.7_{\pm5.0}$ | $99.0_{\pm4.5}$ | $42.2_{\pm4.6}$ |
| FeatProp | $85.7_{\pm0.0}$ | $83.3_{\pm0.0}$ | $100.0_{\pm0.0}$ | $100.0_{\pm0.0}$ | $70.0_{\pm0.0}$ | $60.0_{\pm0.0}$ | $80.0_{\pm0.0}$ | $45.0_{\pm0.0}$ |
| SDCN | $85.7_{\pm9.5}$ | $86.7_{\pm10.5}$ | $93.3_{\pm14.0}$ | $82.5_{\pm8.7}$ | $67.5_{\pm10.4}$ | $72.7_{\pm9.1}$ | $87.5_{\pm10.4}$ | $56.7_{\pm2.9}$ |
| EGAE | $98.6_{\pm4.5}$ | $93.3_{\pm8.6}$ | $86.7_{\pm17.2}$ | $88.8_{\pm4.0}$ | $85.0_{\pm8.5}$ | $76.7_{\pm11.4}$ | $86.0_{\pm9.7}$ | $59.2_{\pm5.8}$ |
| GCC | $100.0_{\pm0.0}$ | $95.0_{\pm8.0}$ | $100.0_{\pm0.0}$ | $87.5_{\pm0.0}$ | $85.0_{\pm7.1}$ | $92.7_{\pm3.8}$ | $100.0_{\pm0.0}$ | $72.5_{\pm3.4}$ |
| DMoN | $90.0_{\pm9.4}$ | $90.0_{\pm10.0}$ | $91.7_{\pm14.8}$ | $87.5_{\pm5.7}$ | $76.0_{\pm9.9}$ | $90.3_{\pm4.6}$ | $97.0_{\pm7.3}$ | $60.2_{\pm6.3}$ |
| RS-GNN | $100.0_{\pm0.0}$ | $90.0_{\pm8.4}$ | $100.0_{\pm0.0}$ | $98.8_{\pm3.9}$ | $82.5_{\pm9.7}$ | $100.0_{\pm0.0}$ | $100.0_{\pm0.0}$ | $54.8_{\pm3.6}$ |

rsults to the RSL problem. In particular, we show that RS-GNN can be applied even when a natural graph structure is not available, but a similarity graph of the nodes is expected to have some degree of homophily. We assume we have access *only* to the node features of our datasets, and not to their graph structures. The baselines select representatives only based on the node features. For RS-GNN, we first create a kNN similarity graph between the nodes and then run the model on the node features and the created graph. The kNN graph is computed once and then fixed during training; we leave further experiments with learning the graph structure as future work. We set the number of neighbors in kNN graphs to 15 in our experiments. Notice that all models have access to the same context.

Once the representatives are selected, we train GCN classifiers on the labels of the selected nodes. For the baselines, we run our classification GCN in two settings: (1) the only edges in the graph are self-loops, (2) the edges in the graph are those from a kNN similarity graph. We call the former a multi-layer perceptron (MLP) and the latter a (kNN-)GCN. We report results for the top baselines that operate on node features *only*.

Since operating without graph structure reduces the signal in the datasets, we allow each model to select $5c$ representatives for the experiments in this section. The results are presented in Table 3. The results show that selecting representatives using RS-GNN performs consistently well on all datasets and provides a boost compared to our baselines on many of the datasets. This confirms that even a similarity graph can still be helpful in improving RSL and that RS-GNN is an effective RSL algorithm for datasets where a graph structure is not available.

We also verified the hardness of the FoN problem (in the absence of a graph) empirically by testing RS-GNN on the FoN problem in Figure 2, where instead of the homophilous graph we use a similarity graph. For sanity check, we also tested KMeans on spectral features in this setting. Both models performed close to random. These results provide empirical evidence for the hardness of RSL in the worst case.

## 7.3 Label Coverage

Once a set of representatives are selected from a dataset, we define a specific class label to be *covered* if at least one of the representatives belongs to that class. We define *label coverage* as the percentage of class

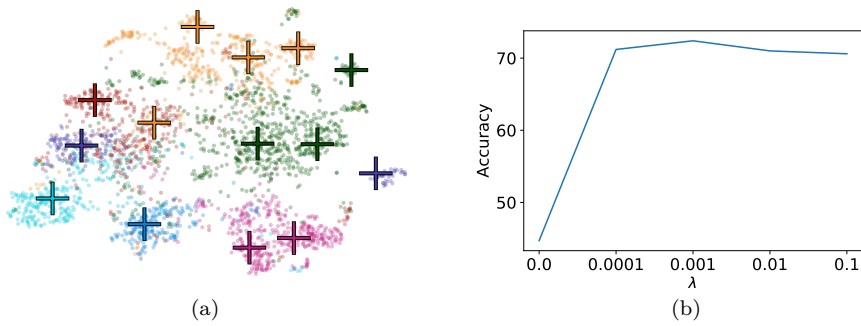

Figure 3: (a) A UMAP visualization of the node embeddings and the selected representatives for Cora (colors represent the class to which the nodes/representatives belong), (b) RS-GNN results on Cora for different values of $\lambda$.

labels that are covered by the selected representatives. In Table 4, we report the label coverage results for the baselines and our model when selecting $k = 2c$ representatives. RS-GNN performs well in terms of selecting points that cover all labels, with a coverage of 100.0% on four of the datasets. We observe an interesting phenomenon on the Arxiv dataset where (unlike other datasets) many baselines outperform RS-GNN in terms of label coverage (especially GCC), but RS-GNN shows a higher accuracy compared to the baselines. We believe this is due to the high label imbalance in this dataset (the largest class has 27321 examples and the smallest has 29 examples). It is also worth noting that Arxiv has the largest number of nodes, edges, and classes and the lowest number of base features among our datasets. Future work can extend our work for optimizing macro accuracy across classes.

## 7.4 Visualization

In Figure 3(a), we use UMAP McInnes et al. (2018) to visualize the learned embeddings and the selected representatives of RS-GNN for the Cora dataset. The colors in the plot represent the class to which the node or the representative belongs. According to the visualization. the nodes from each class have formed one or more dense clusters and our model has selected a representative from (almost all of) these dense regions of points. More visualizations can be found in supplementary material.

## 7.5 Time Complexity and Memory Usage

Let $m$ represent the number of nodes, $d$ represent the average degree of nodes, $n$ represent the number of features (and, for simplicity, the embedding dimension), and $k$ represent the number of representatives. The time complexity of each epoch in Algorithm 1 is goverened by $O(mnd)$ for computing graph convolusions, $O(mn^2)$ for computing node projections and bilinear functions, and $O(mnk)$ for assigning nodes to representatives. Overall, this gives a time complexity of $O(mn(d + n + k))$ for each epoch. For large datasets where a large number of representatives is needed (i.e. $k > n$ and $k > d$), the time complexity becomes $O(mnk)$. The memory complexity is $O(mk)$ for constructing and storing the matrix of distances from each node to its representatives.

When $m$ and $k$ are both large, Algorithm 1 may exhaust the accelerator memory due to its $O(mk)$ complexity. To reduce the memory usage and allow for applying RS-GNN to such settings, we modify Algorithm 1 to Algorithm 2 (in the Appendix). The main modifications include: 1- as is common in the GNN scalability literature (see, e.g., Frasca et al. (2020)), we replace the GCN modules with a SGC module (Wu et al., 2019a) and pre-compute the graph convolutions $\boldsymbol{F}$ for the original graph, 2- we create corrupted graphs $nCorrupt$ times and pre-compute the graph convolutions $\boldsymbol{F'}$ for the corrupted graphs, 3- at the beginning of each epoch, we randomly select a subset $\boldsymbol{F''}$ of $\boldsymbol{F'}$ to make the size match that of $\boldsymbol{F}$, 4- we batch the data and compute the loss and gradients for each batch separately and then aggregate the gradients and update the parameters. Assuming the batch size is $b$, the memory complexity for Algorithm 2 reduces to $O(bk)$ as each batch can be

Table 5: Ablation study results for RS-GNN.

| Ablation | Cora | CiteSeer | PubMed | Photos | PC | CS | Physics | Avg. |
|---|---|---|---|---|---|---|---|---|
| NoNorm | $61.6_{\pm 7.7}$ | $41.5_{\pm 3.5}$ | $59.6_{\pm 4.7}$ | $82.5_{\pm 2.1}$ | $71.3_{\pm 2.8}$ | $86.6_{\pm 2.4}$ | $90.8_{\pm 1.5}$ | 70.6 |
| ConstNorm | $59.3_{\pm 6.9}$ | $44.3_{\pm 4.2}$ | $61.7_{\pm 3.9}$ | $84.2_{\pm 1.4}$ | $75.0_{\pm 3.1}$ | $86.7_{\pm 1.3}$ | $90.5_{\pm 2.1}$ | 71.7 |
| Full Model | $72.4_{\pm 3.0}$ | $54.7_{\pm 3.9}$ | $65.8_{\pm 3.0}$ | $86.3_{\pm 1.4}$ | $74.3_{\pm 1.7}$ | $89.3_{\pm 0.8}$ | $90.0_{\pm 2.6}$ | 76.1 |

computed separately. Moreover, Algorithm 2 is also amenable to parallelization on multiple accelerators by distributing the batches across the accelerators.

Note that computing the loss separately for each batch corresponds to approximating $s$ and $\mu$ with the batch data as opposed to the entire data, but the approximation is expected to be close to the true value if the batch sizes are large enough. To this end, the batch size can be set to the largest number that does not exhaust the memory. We tested the Algorithm 2 version of RS-GNN on the Arxiv dataset when setting *nCorrupt* to 10 and $b = {}^m/_8$ (distributing over 8 accelerators). In terms of performance, we obtained an accuracy of $52.9 \pm 1.6$ (compared to $52.6 \pm 1.2$ for the original algorithm) showing that the approximations do not result in a performance degradation.

### 7.6 Ablation Study: CenterNorm and the Value of $\lambda$

We conduct an ablation study to verify the role of CenterNorm. To ablate CenterNorm, we run our model under two settings: (1) we do not normalize (we refer to this as "NoNorm"), and (2) we divide all the values in the embedding matrix by a constant number corresponding to the mean of the $\ell_2$-norms of the embeddings (we refer to this as "ConstNorm"). According to the results in Table 5, our model benefits from CenterNorm and CenterNorm is more effective than ConstNorm because the per-node $\ell_2$ normalization of CenterNorm helps bring the nodes within one cluster closer to each other which helps in identifying clusters and selecting representatives.

To verify the sensitivity of RS-GNN to the value of $\lambda$ used in equation 1, we ran RS-GNN on Cora with different values for $\lambda$. The results are presented in Figure 3(b). When $\lambda = 0$, the representatives will not receive gradients and hence the model ends up selecting random representatives. Therefore, the accuracy is quite low. For non-zero values of $\lambda$, we observe that the model is not highly sensitive to the value of $\lambda$ and achieves good results for values in a large range. The model reaches its highest performance around $\lambda = 0.001$, and then the performance starts to slightly decrease for larger values of $\lambda$.

## 8 Conclusion

In this paper, we studied the representative selection problem for attributed graphs theoretically and empirically. In the absence of a graph structure, we proved new hardness results showing it is impossible to provide a polynomial-time algorithm for this problem with an approximation within any reasonable factor, unless the exponential time hypothesis fails. The hardness result explains the significant gap between the accuracy of models trained on optimal representatives and the widely-used surrogate functions for RS problems, and, in turn, justifies new techniques to solve this problem. We then showed the existence of a homophilous graph structure can turn a hard representative selection problem into one that can be effectively solved, when using an appropriate modelling technique. In light of this result we proposed RS-GNN to optimize the task of representative selection for attributed graphs via graph neural networks, and showed its effectiveness on a suite of different datasets and tasks.

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

## A    More results

**Selecting more representatives:**   We already presented results for the case where we select $2c$ representatives from each dataset, where $c$ is the number of classes. Here, we present more results for the case where we select $5c$ representatives from each class. We limit our baselines to the ones that were either more competitive/informative or took less time to run. The results are in Table 6. We can observe that the performance for all models improves when more representatives are selected. We can also observe that the gap between the models shrinks when more representatives are selected (even the random baseline now shows a competitive performance). Nevertheless, RS-GNN shows a similar trend as when we selected $2c$ representatives and outperforms the baselines when averaged across datasets.

**Visualization:**   In the main text, we visualized the learned embeddings and the selected representatives for the Cora dataset. In Figure 4(a), we provide a visualization for Citeseer. Similar behaviour as that of Cora can be observed for Citeseer, where node embeddings form clusters and a representative is selected from each of the dense cliusters.

**Attributed Graph Clustering:**   RS-GNN can also be used for attributed graph clustering: we cluster the nodes in each dataset into $c$ groups (recall that $c$ is the number of classes) by assigning each node to its closest representative. While our main motivation is RSL, for completeness we show the performance of our model for attributed graph clustering as well. We note that different works on attributed graph clustering use different settings that are not directly comparable (e.g., some works use the same hyperparameters for all datasets, whereas some other works optimize the hyperparameters for each dataset and report the best test performance). Since our setting is similar to that of Tsitsulin et al. (2023) and the results for many models have been provided in that work, we compare RS-GNN against the model proposed in (Tsitsulin et al., 2023) and their baselines. This includes KMeans, SBM (this works by estimating (Peixoto, 2014) a

Table 6: For each dataset, each algorithm selects $5c$ representatives. Then, we train a GCN model on the labels of the selected representatives. The reported metric is the test accuracy of the GCN models. Surrogate function selectors mark in blue use DGI embeddings.

| Selector | Cora | CiteSeer | Pubmed | Photos | PC | CS | Physics | Arxiv | Avg. |
|---|---|---|---|---|---|---|---|---|---|
| Random | $66.3_{\pm4.6}$ | $47.3_{\pm6.6}$ | $62.0_{\pm8.3}$ | $84.0_{\pm3.5}$ | $76.8_{\pm3.4}$ | $83.9_{\pm2.3}$ | $84.4_{\pm5.8}$ | $54.6_{\pm1.2}$ | 69.9 |
| Popular | $64.1_{\pm0.6}$ | $43.2_{\pm1.4}$ | $63.4_{\pm0.9}$ | $43.9_{\pm1.1}$ | $54.1_{\pm1.4}$ | $78.6_{\pm0.5}$ | $50.5_{\pm0.0}$ | $42.6_{\pm1.4}$ | 55.0 |
| KMedoid | $62.2_{\pm1.3}$ | $42.1_{\pm3.1}$ | $60.8_{\pm0.3}$ | $78.8_{\pm0.6}$ | $74.4_{\pm1.2}$ | $63.9_{\pm1.7}$ | $64.3_{\pm0.6}$ | $50.6_{\pm0.5}$ | 62.1 |
| KMeans | $66.7_{\pm1.1}$ | $53.8_{\pm1.3}$ | $71.6_{\pm0.5}$ | $84.8_{\pm1.0}$ | $81.4_{\pm0.7}$ | $76.7_{\pm1.0}$ | $80.6_{\pm0.3}$ | $56.8_{\pm0.3}$ | 71.6 |
| MC-Cos | $71.3_{\pm2.8}$ | $59.0_{\pm2.8}$ | $64.8_{\pm0.3}$ | $87.5_{\pm0.4}$ | $78.3_{\pm0.7}$ | $88.4_{\pm0.2}$ | $92.4_{\pm0.4}$ | $56.2_{\pm0.5}$ | 74.7 |
| KMeans | $74.9_{\pm3.1}$ | $54.3_{\pm4.8}$ | $69.3_{\pm3.2}$ | $89.4_{\pm1.3}$ | $82.0_{\pm1.4}$ | $89.2_{\pm0.8}$ | $92.1_{\pm0.7}$ | $55.1_{\pm1.1}$ | 75.8 |
| MC-Cos | $75.8_{\pm1.7}$ | $59.8_{\pm2.5}$ | $70.6_{\pm2.4}$ | $90.0_{\pm1.2}$ | $82.3_{\pm1.1}$ | $89.4_{\pm0.8}$ | $86.1_{\pm1.9}$ | $55.8_{\pm0.7}$ | 76.2 |
| MinCUT | $66.8_{\pm4.3}$ | $48.7_{\pm6.6}$ | $69.0_{\pm4.2}$ | $16.7_{\pm8.1}$ | $16.5_{\pm10.2}$ | $88.6_{\pm1.2}$ | $92.3_{\pm1.0}$ | $55.3_{\pm7.5}$ | 56.7 |
| DMoN | $70.1_{\pm3.8}$ | $54.9_{\pm4.6}$ | $70.2_{\pm4.0}$ | $86.7_{\pm2.4}$ | $74.8_{\pm5.8}$ | $89.7_{\pm0.7}$ | $90.9_{\pm1.8}$ | $57.1_{\pm1.0}$ | 74.3 |
| RS-GNN | $77.3_{\pm1.9}$ | $62.7_{\pm2.3}$ | $68.7_{\pm2.4}$ | $90.6_{\pm0.5}$ | $83.0_{\pm1.6}$ | $90.1_{\pm0.6}$ | $92.4_{\pm0.8}$ | $56.6_{\pm1.2}$ | 77.7 |

Table 7: Normalized mutual information (NMI) scores between the ground-truth labels of nodes and their cluster assignments. We take the results of the baselines from Tsitsulin et al. (2023). Bold numbers indicate the winners. We use — to indicate that the results were not reported (either because the model did not converge, or because the model did not scale to the dataset, or because the model was not tested on the dataset).

| Selector | Context | Cora | CiteSeer | Pubmed | Photos | PC | CS | Physics | Avg. |
|---|---|---|---|---|---|---|---|---|---|
| KMeans | $\mathbf{X}$ | 18.5 | 24.5 | 19.4 | 28.8 | 21.1 | 35.7 | 30.6 | 25.5 |
| SBM | $\mathbf{A}$ | 36.2 | 15.3 | 16.4 | 59.3 | 48.4 | 58.0 | 45.4 | 39.9 |
| MinCut | $\mathbf{X}, \mathbf{A}$ | 35.8 | 25.9 | 25.4 | — | — | 64.6 | 48.3 | — |
| AGC | $\mathbf{X}, \mathbf{A}$ | 34.1 | 25.5 | 18.2 | 59.0 | **51.3** | 43.3 | — | — |
| DAEGC | $\mathbf{X}, \mathbf{A}$ | 8.3 | 4.3 | 4.4 | 47.6 | 42.5 | 36.3 | — | — |
| SDCN | $\mathbf{X}, \mathbf{A}$ | 27.9 | 31.4 | 19.5 | 41.7 | 24.9 | 59.3 | 50.4 | 36.4 |
| NOCD | $\mathbf{X}, \mathbf{A}$ | 46.3 | 20.0 | 25.5 | 62.3 | 44.8 | 70.5 | 51.9 | 45.9 |
| DMoN | $\mathbf{X}, \mathbf{A}$ | 48.8 | 33.7 | **29.8** | **63.3** | 49.3 | 69.1 | 51.9 | 49.4 |
| RS-GNN | $\mathbf{X}, \mathbf{A}$ | $\mathbf{55.4}_{\pm0.8}$ | $41.3_{\pm1.0}$ | $26.1_{\pm3.6}$ | $58.3_{\pm1.6}$ | $50.1_{\pm0.9}$ | $\mathbf{75.8}_{\pm1.2}$ | $\mathbf{56.9}_{\pm3.3}$ | **52.0** |

constrained Stochastic Block Model (Snijders & Nowicki, 1997) with given number of $k$ clusters), MinCut, AGC (Zhang et al., 2019), DAEGC (Wang et al., 2019), SDCN, NOCD (Shchur & Günnemann, 2019), and DMoN (Tsitsulin et al., 2023).

Table 7 shows a comparison of RS-GNN with the baselines in terms of the normalized mutual information (NMI) score (an established score for measuring and comparing clustering algorithms) between the cluster assignments and the node labels. From the results, we can observe that RS-GNN also shows a good performance for attributed graph clustering.

# B  CenterNorm Motivation

With the joint loss function used in the main text, the model can trivially reduce $\mathcal{L}_{\mathsf{SEL}}$ by making the values in the embedding matrix $\boldsymbol{H}$ arbitrarily small. That is because multiplying a small constant to $\boldsymbol{H}$ may not change $\mathcal{L}_{\mathsf{EMB}}$ substantially, but it can make the distances between the nodes arbitrarily small, resulting in a low value for $\mathcal{L}_{\mathsf{SEL}}$ even for a random representative embedding matrix $\boldsymbol{R}$.

One way to avoid the aforementioned problem is by normalizing the embedding matrix $\boldsymbol{H}$ before using it for selection. However, one should be careful about the choice of the normalization to avoid losing useful

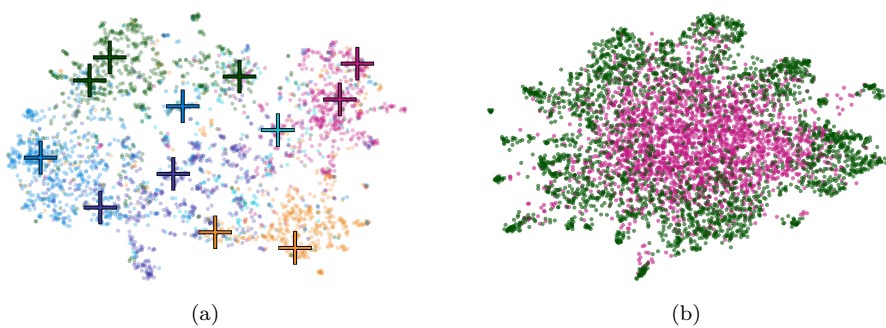

Figure 4: (a) A UMAP visualization of the node embeddings and the selected representatives for Citeseer. The colors represent the class to which the node or the representative belongs, (b) A UMAP plot of the DGI node embeddings for the nodes in the original graph of Cora (green) and the nodes in a corrupted Cora graph (red).

information. Before explaining how we normalize the embeddings, we describe a property of DGI embeddings that motivates our normalization.

Figure 4(b) shows a UMAP plot of the DGI embeddings $H$ for the nodes in the original graph of Cora and the embeddings $H'$ for the nodes in a corrupted Cora graph (red). The $H'$ embeddings form a large cluster that is mostly in between the $H$ embeddings and the $H$ embeddings form small size clusters that are placed around the large cluster of $H'$. To understand why this happens, notice that when we shuffle the node features for creating corrupted graphs for DGI training, the node features of the neighbors of each node are a random subsample of the node features in the graph. Therefore, the GNN aggregation function applied on the projected node embeddings makes the embeddings go toward the mean of projected embeddings. This makes the corrupted node embeddings $H'$ form a large cluster in the middle and the embeddings $H$ be placed outside and around this cluster.

Besides visual inspection, we also cluster the node embeddings $H$ for Cora into 7 clusters using KMeans (7 is the number of classes in Cora; this provides good clusters with a normalized mutual information of 55.95 with the node labels). Then we compute the distance between each cluster center and the mean of these centers and obtain the following seven distances: 1.4, 1.2, 1.3, 1.2, 0.6, 1.4, 1.5. All cluster centers are at a good distance from the mean and, with the exception of one cluster, they are at a similar distance from the mean. We then subtract the mean and compute the angle between the cluster centers. We observe that the minimum angle between two cluster centers is 60.1 degrees and the average angle is 76.6 degrees.

The above analysis motivates the CenterNorm normalization outlined in the main text. Furthermore, the analysis shows that DGI is a good candidate for RSL as it groups data points into small-sized dense clusters in the latent space, thus an RSL algorithm can select representatives from each of the dense clusters.

## C   Proof of the Theorem

**Theorem 2.** *There is no polynomial-time representative selection algorithm for FoN with an approximation factor better than $\omega(n^{-1/\mathrm{poly}\log\log n})$, unless the ETH fails.*

We start with two lemmas before proceeding to prove this result. We use $(i, j)$ to represent a data point in the FoN problem with value 1 on features $i$ and $j$.

**Lemma 1.** *Let $\mathcal{S}$ be the set of data points selected by an RSL algorithm. Let $(i, j)$ be a data point such that for all $t$ we have $(i, t) \notin \mathcal{S}$, or for all $t$ we have $(t, j) \notin \mathcal{S}$. Then the label of $(i, j)$ is independent of the labels of $\mathcal{S}$.*

*Proof.* Let $\mathcal{S}$ be the set of data points selected by an RSL algorithm. Let $(i, j)$ be a data point such that for all $t$ we have $(i, t) \notin \mathcal{S}$. This means that the labels of data points in $\mathcal{S}$ are independent of the type of feature

$i$. Recall that the type of feature $i$ is chosen independently and uniformly at random. Hence, conditioned on the labels in $\mathcal{S}$, the label of $(i, j)$ is 0 with probability $1/2$ and 1 with probability $1/2$. Similar argument holds when for all $t$ we have $(t, j) \notin \mathcal{S}$. □

**Lemma 2.** *Let $(i_0, i_1), (i_1, i_2), (i_2, i_3), \ldots, (i_{l-1}, i_l)$ be a sequence of data points such that for all $t \in \{1, \ldots, l\}$ we have $(i_{t-1}, i_t) \in \mathcal{S}$. Given the labels of the data points in $\mathcal{S}$ we can infer the label of $(i_0, i_l)$.*

*Proof.* Consider two data points $(i, j)$ and $(j, t)$. If the labels of both data points are 1, then the features $i$, $j$ and $t$ have the same type. Hence, the label of $(i, t)$ is 1 too. If the labels of both of them are 0, then the type of features $i$ and $j$ are different, and the type of features $j$ and $t$ are different. Hence, the type of features $i$ and $t$ are the same, which means the label of $(i, t)$ is 1. A similar argument shows that if either $(i, j)$ or $(j, t)$ has label 1 and the other has label 0, then the label of $(i, t)$ is 0. Thus, knowing the labels of $(i, j)$ and $(j, t)$ determines the label of $(i, t)$. Applying this inductively proves the lemma. □

*Proof of Theorem 1.* The proof goes by reducing the densest $k$-subgraph problem to FoN. In the densest $k$-subgraph problem, we have an unweighted graph $\mathbb{G}$, and the goal is to find a subgraph of $\mathbb{G}$ with $k$ vertices and the maximum number of edges. We say an algorithm is an $\alpha$-approximation algorithm for the densest $k$-subgraph problem if it returns a subgraph with $k$ vertices where the number of edges is at least $\alpha$ times that of the densest $k$-subgraph. It is known that there is no $\omega(n^{-1/\text{poly} \log \log n})$-approximation polynomial-time algorithm for the densest $k$-subgraph problem unless ETH fails Manurangsi (2017).

Next, we show how to transform an input of the densest $k$-subgraph problem to an input of FoN, and then show how to transform an approximate solution for FoN to an approximate solution for the densest $k$-subgraph problem while only increasing the approximation factor by a constant. Therefore an $\omega(n^{-1/\text{poly} \log \log n})$-approximation polynomial-time algorithm for the FoN implies an $\omega(n^{-1/\text{poly} \log \log n})$-approximation polynomial-time algorithm for the densest $k$-subgraph problem, which does not exist unless ETH fails.

Let $\mathbb{G} = (V, E)$ be an input to the densest $k$-subgraph problem.[6] For each vertex in $\mathbb{G}$ we define a feature and for each edge in $\mathbb{G}$ we construct a data point. For each data point corresponding to an edge $(u, v)$, the value of the features corresponding to vertices $u$ and $v$ are 1 and the value of all other features are 0. As defined in the FoN problem the type (red or blue) of each feature is chosen independently and uniformly at random.

Let $\mathbb{H} = (V_{\mathbb{H}}, E_{\mathbb{H}})$ be a densest $k$-subgraph of $\mathbb{G}$ and let $\mathbb{F}$ be a maximal spanning forest of $\mathbb{H}$. Note that since there is no cycle in $\mathbb{F}$, the number of edges in $\mathbb{F}$ is at most $k - 1$. Moreover, since $\mathbb{F}$ is a maximal forest of $\mathbb{H}$, for each edge $e$ in $\mathbb{H}$, there is a path between the endpoints of $e$ in $\mathbb{F}$ (otherwise we could add $e$ to $\mathbb{F}$). Hence, if we query the data points corresponding to the edges of $\mathbb{F}$, by Lemma 2, we can determine the label of all the edges in $\mathbb{H}$, which is an $\frac{|E_{\mathbb{H}}|}{|E|}$ fraction of all data points. This gives us a solution with $\overline{\text{Acc}} \geq \Omega\left(\frac{|E_{\mathbb{H}}|}{|E|}\right)$.

Let $\mathcal{S}$ be the set of data points selected by an $\alpha$-approximation RSL algorithm, and $V_{\mathcal{S}}$ be the set of vertices adjacent to the edges corresponding to the data points in $\mathcal{S}$. By Lemma 1, if the edge corresponding to a data point has one (or two) endpoints in $V \setminus V_{\mathcal{S}}$, then the label of that data point is independent of the labels of $\mathcal{S}$. Hence, the number of data points whose label is not independent of the labels in $\mathcal{S}$ is at most the number of edges induced by $V_{\mathcal{S}}$. We denote this edge set by $E_{\mathcal{S}}$. Recall that $\mathcal{S}$ is an $\alpha$-approximate solution, i.e., $|E_{\mathcal{S}}| = \Omega(\alpha|E_{\mathbb{H}}|)$. On the other hand, $|\mathcal{S}| \leq k$ and hence $|V_{\mathcal{S}}| \leq 2k$. One can decompose the induced subgraph of $V_{\mathcal{S}}$ into $\binom{4}{2} = 6$ subgraphs each with $k$ vertices, and pick the one with the maximum number of edges. This gives an $\Omega(\alpha)$-approximate solution to the densest $k$-subgraph problem. □

# D  Implementation Details

**Baselines:** For KMeans, we select the closest node to each cluster center as a representative. For FFS and MaxCover, we select representatives sequentially. In FFS, the next representative is the node farthest away (by Euclidean distance) from the closest representative in the current set. In MaxCover, the next representative is the node that increases the coverage of the non-selected nodes the most. Note that the

---

[6]Note that graph $\mathbb{G}$ is not an attributed graph, rather a simple graph which is an input to the densest $k$-subgraph problem.

sequential nature of FFS and MC makes them less amenable to parallelization. Also note that when we run surrogate function baselines using DGI embeddings as context, their selections are informed by both node features and the graph structure. For the spectral embeddings, we used the top 100 singular vectors. For DMoN and MinCut models, we compute cluster centers by averaging the node embeddings with respect to the (hard) cluster assignments, and then select the closest point to each cluster center as a representative.

We implemented our model and the baselines in Jax/Flax (Bradbury et al., 2018; Heek et al., 2020) and used the Jraph library (Godwin* et al., 2020) for our GNN operations. Our experiments were done on a TPU v2 for all datasets except for the Arxiv dataset where we used a TPU v3 as the experiments with the Arxiv dataset require more memory. For our DGI model, we used a single-layer GCN model with SeLU activations (Klambauer et al., 2017). We set the learning rate to 0.001 and optimized all model parameters (including the DGI parameters and the center parameters) jointly. For the experiments that had access to the original graph structure, we set the DGI hidden dimension to 512 for all datasets except for the Arxiv dataset where we set it to 256 to reduce memory usage. For the experiments with no access to the original graph structure, we set the DGI hidden dimension to 128 as there exists less signal in this case. We trained the DGI models for 2000 epochs both for our model and the baselines. For

---

**Algorithm 2** Memory-Efficient RS-GNN.

**Input:** $\mathcal{G} = (\mathcal{V}, \boldsymbol{A}, \boldsymbol{X})$, $k$

1: Initialize $\boldsymbol{R}$, $\boldsymbol{\Theta}$, and $\boldsymbol{U}$
2: $\boldsymbol{F} = \mathsf{GC}(G)$, $\boldsymbol{F}' = []$
3: **for** i=1 **to** nCorrupt **do**
4:     $\mathcal{G}' = (\mathcal{V}, \boldsymbol{A}, \mathsf{shuffle}(\boldsymbol{X}))$
5:     $\boldsymbol{F}' = \mathsf{concat}(\boldsymbol{F}', \mathsf{GC}(\mathcal{G}'))$
6: **for** epoch=1 **to** #epochs **do**
7:     $\boldsymbol{F}'' = \mathsf{subsample}(\boldsymbol{F}', \mathsf{len}(\boldsymbol{F}))$
8:     $\nabla = \boldsymbol{0}$
9:     **for** $\boldsymbol{F}^{(b)}, \boldsymbol{F}^{(b'')} \in \mathsf{batch}(\boldsymbol{F}, \boldsymbol{F}'')$ **do**
10:       $\boldsymbol{H} = \boldsymbol{W}\boldsymbol{F}^{(b)}$ ,    $\boldsymbol{H}' = \boldsymbol{W}\boldsymbol{F}^{(b'')}$
11:       Compute $\mathcal{L}_{\mathsf{EMB}}$ based on $\boldsymbol{H}$ and $\boldsymbol{H}'$
12:       $\boldsymbol{\mu} = \frac{1}{n}\sum_i \boldsymbol{H}_i, \boldsymbol{\zeta} = \|\boldsymbol{H} - \boldsymbol{\mu}\|$
13:       $\tilde{\boldsymbol{H}} = \mathsf{CenterNorm}(\boldsymbol{H}) = (\boldsymbol{H} - \boldsymbol{\mu})/\boldsymbol{\zeta}$
14:       $\mathcal{L}_{\mathsf{SEL}} = \sum_i \min_j \mathsf{Dist}(\tilde{\boldsymbol{H}}_i, \boldsymbol{R}_j)$
15:       $\mathcal{L} = \mathcal{L}_{\mathsf{EMB}} + \lambda\mathcal{L}_{\mathsf{SEL}}$
16:       Compute gradients for $\mathcal{L}$ and add to $\nabla$
17:     Update parameters based on $\nabla$
18: Let $\hat{\boldsymbol{R}}$ and $\hat{\boldsymbol{H}}$ be the representative and normalized node embeddings with minimum $\mathcal{L}$ during training.
19: **for** j=1 **to** k **do**
20:     The $j^{\mathrm{th}}$ representative = $\mathsf{argmin}_i\mathsf{Dist}(\hat{\boldsymbol{H}}_i, \hat{\boldsymbol{R}}_j)$

---

KMeans and KMedoid, we used the implementation in scikit-learn (Pedregosa et al., 2011) and scikit-learn-extra[7] respectively. To reduce the quadratic time complexity of MaxCover, we apply MaxCover on a k-nearest neighbors similarity graph in the input features/embeddings as opposed to the full graph. We used different hyperparameters for the RBF kernel (in the case of MaxCover with RBF similarities) and kNN and reported the values that resulted in the best overall accuracy across models. For MinCUT and DMoN, we used the implementation from the DMoN paper. For our model, we set $\lambda$ in the main loss function to 0.001 for all datasets. Also, for the experiments where a graph structure is not provided as input, to create a kNN graph we connect each node to its closest 15 nodes for all the datasets. For the one-shot graph active learning models, unfortunately we did not find the code to be able to test the models in our setting. Therefore, we re-implemented FeatProp (Wu et al., 2019b) and included the results of our implementation in the experiments. For SDCN, EGAE, and GCC, we used the public codes released by the authors to select representatives.

For the classification GCN model, we used a two-layer GCN model with PReLU activations (He et al., 2015) and with a hidden dimension of 32. We added a dropout layer after the first layer with a drop rate of 0.5. The weight decay was set to $5e^{-4}$. The GCN is trained based on the nodes in the selected set $\mathcal{S}$ of representatives. We randomly split the remaining nodes in $(\mathcal{V} - \mathcal{S})$ into validation and test sets by selecting 500 nodes for validation and the rest for testing.

We ran all the experiments 20 times (except for Arxiv where we ran it 10 times) with different random seeds and reported the mean and standard deviation of the runs. Our code will be released upon the acceptance of the paper.

---

[7]https://github.com/scikit-learn-contrib/scikit-learn-extra

Table 8: Dataset statistics.

| Dataset | Nodes | Edges | Features | Classes |
|---------|-------|-------|----------|---------|
| Cora | 2,708 | 5,278 | 1433 | 7 |
| Citeseer | 3,312 | 4,536 | 3703 | 6 |
| Pubmed | 19,717 | 44,324 | 500 | 3 |
| Amazon Photo | 7,650 | 119,081 | 745 | 8 |
| Amazon PC | 13,752 | 245,861 | 767 | 10 |
| Coauthor CS | 18,333 | 81,894 | 6,805 | 15 |
| Coauthor PHY | 34,493 | 247,962 | 8,415 | 5 |
| OGBN-Arxiv | 169,343 | 1,157,799 | 128 | 40 |

## E  Datasets

We used eight established benchmarks in the GNN literature. A summary of our dataset statistics are provided in Table 8. The first three datasets are Cora, Citeseer, and Pubmed (Sen et al., 2008; Hu et al., 2020). These datasets are citation networks in which nodes represent papers, edges represent citations, features are bag-of-word abstracts, and the labels represent paper topics. The next two datasets are Amazon Photo and Amazon PC (Shchur et al., 2018). These two datasets correspond to photo and computers subgraphs of the Amazon copurchase graph. In these graphs, the nodes represent goods with an edge between two nodes representing that they have been frequently purchased together. Node features are bag-of-word reviews and class labels are product categories. The next two datasets are Coauthor CS and Coauthor Physics (Shchur et al., 2018). These are co-authorship networks for the computer science and physics fields based on the Microsoft Academic Graph respectively. The nodes in these two datasets represent authors, edges represent co-authorship, node features are a collection of paper keywords from author's papers, and he class labels are the most common fields of study. Our last dataset is OGBN-Arxiv (Hu et al., 2020) which is also a citation dataset similar to Cora, Citeseer, and Pubmed, but orders of magnitude bigger than the three. The features in this dataset are average word embeddings of the paper abstracts.

## F  Extra Related Work

In what follows, we continue our discussion of related work and provide description of categories of work that are also relevant to our work.

**Feature selection:** RS and feature selection are transposed views of a similar problem when it comes to compressing or summarizing datasets. Both have been studied extensively via *filter* and *wrapper* methods: an evaluation based on final task performance or on some proxy metric such as correlation, redundancy, coverage (or more general submodular functions), or the distance of selected entities (Lee & Chung, 2000; Pan et al., 2005; Bolón-Canedo et al., 2015; Bateni et al., 2018) as well as their mutual information or correlation with the prediction labels (Ding & Peng, 2003; Novovicová et al., 2007). While these methods tackle the diversity of the sample set (Abbassi et al., 2013; Indyk et al., 2014; Zadeh et al., 2017; Bhaskara et al., 2016), there has also been extensive attention on taking fairness constraints into account as well (Kou et al., 2021; Roh et al., 2021; Lee et al., 2021b; Shekhar et al., 2021; Aumüller et al., 2021).

**Supervised data subset selection:** Given a large dataset of labeled training examples, a class of RS models aim at selecting a small representative set from the dataset to reduce training time without substantially sacrificing model accuracy (see, e.g., Killamsetty et al. (2021b;a); Wei et al. (2015); Kaushal et al. (2019); Durga et al. (2021); Paul et al. (2021); Mirzasoleiman et al. (2020)). For example, Killamsetty et al. (2021b) aim at selecting a set of training data points such that a model trained on these examples generalizes well to the validation set and Mirzasoleiman et al. (2020) aim at selecting a set of training data points whose gradients approximate the gradient of the full dataset. These models have been also applied to mini-batch active learning where a small set of labeled data points are assumed to be initially provided and then the next batches are selected based on pseudo-labels predicted by the model trained on the data available so

far. While these models assume the data labels are available when selecting representatives, in this paper we assume no labels are provided as input.

**Graph pooling:** A technique commonly used in graph representation learning (especially for learning a representation for the entire graph) is graph pooling (Liu et al., 2022a), where the nodes of the graph are iteratively coarsened into "super-nodes". The parameters for the pooling operation are trained with the rest of the model parameters either to minimize a supervised loss (e.g., graph classification) or an unsupervised loss (e.g., graph reconstruction) (Gao & Ji, 2019; Ge et al., 2021). Graph pooling techniques can be classified into two categories: 1- clustering pooling (these are in the same vein as the graph clustering algorithms discussed earlier) and 2- node drop pooling. Clustering pooling approaches (Ying et al., 2018; Lee et al., 2021a; Yuan & Ji, 2020; Ahmadi, 2020; Bianchi et al., 2020; Liu et al., 2021) employ a differentiable graph clustering algorithm and consider each cluster to be a super-node; these approaches can be re-purposed for RS by selecting the node closest in the latent space to each super-node as a representative. Node drop clustering approaches (Gao & Ji, 2019; Lee et al., 2019; Pang et al., 2021; Zhang et al., 2020; Gao et al., 2021) operate by dropping unimportant nodes and retaining the important nodes as the super-nodes. Graph pooling approaches have been mostly developed for smaller-sized graphs such as molecules that exhibit specific properties. For example, many of these approaches employ a GNN that assigns importance scores to each node, and then select the top-k most similar nodes. Such an approach assigns similar importance scores to highly similar nodes and results in sampling only from some parts of the graph. While this might be a reasonable approach for molecule classification tasks (as it makes the model focus on a few important sub-structures), it may not select a subset of the nodes that cover the entire graph (which is the desired property in our work). Nevertheless, in our experiments, we compare against several graph pooling approaches from both categories, both for RS and clustering tasks.

# G   Limitations

We identify the following limitations with our current work:

- Both our model and baselines optimize for micro-average classification accuracy; optimizing for macro-average classification accuracy may require extra terms in the loss function or architectural modifications.
- While optimizing for micro-average accuracy is common in various domains, it raises the risk of being unfair to smaller sub-populations by not selecting any representatives from them. One must be cautious when using our model or any other model that optimizes for micro-average accuracy in applications when such a fairness is important.

