# OpenReview forum: "Tackling Provably Hard Representative Selection via Graph Neural Networks"
_TMLR — Accepted by TMLR_

### Review · Reviewer_Mgut · 2023-05-22

**Summary Of Contributions:**

The authors study the problem of Representative Selection for Learning (RSL), which aims to learn a subset of points (in unsupervised setting) such that the downstream model trained on this subset of points (in supervised setting) has performance as good as possible. The authors first provide a hardness result of the problem, then introduce the homophilous graph as the side-information. Finally, the authors propose RS-GNN model for RSL and demonstrate strong empirical performance against baseline methods on eight benchmark datasets.

**Audience:**

Yes

**Broader Impact Concerns:**

I do not aware of significant ethical or social impact concerns.

**Claims And Evidence:**

Yes

**Requested Changes:**

1. Discuss the relation to cSBM [1] and compare with methods that works well or achieves information limits for cSBM in unsupervised setting.

2. Conduct experiments on synthetic datasets constructed as described in Section 5. Especially test RS-GNN without graph (induced by node features) on the hard example constructed therein and cSBM.

3. Can we say anything about the GFoN problem in theory?

4. Instead of termed graph as ``side-information``, I think it should be a new problem termed like RSL with graph or something similar. If the authors insist on calling the graph ``side-information’’, I feel the authors should justify that the graph is indeed not the major source of information.

5. One way of resolving bullet 4 above is to conduct experiments on standard classification benchmarks using RS-GNN with node feature induced graph. If RS-GNN can still outperform baseline methods in standard classification benchmarks, then I guess calling the graph a ``side-information`` sounds more reasonable.


**Strengths And Weaknesses:**

## Strengths

(+) Interesting hardness results for RSL.

(+) Strong empirical performance of the proposed RS-GNN model.

## Weaknesses

(-) Lack of discussion and comparison to the relevant contextual stochastic block model literature.

(-) The introduced side-information (homophilous graph) makes the RSL problem a different problem, and the hardness of this problem is unclear in this paper.

(-) The hardness results kind of contradict the claim that RS-GNN can work without graphs (i.e., case of graphs induced from node features).

(-) The experiments heavily rely on the graph models and graph datasets (even if the authors exclude the graph in some experiment). It should also be tested on standard classification benchmarks if graph is really a ``side-information``, especially RS-GNN is claimed to work when even when the graph does not exist.

## Detail comments

While I like the methodology of the paper and the interesting hardness result of RSL, I think the flow of the paper is problematic and misleading. Firstly, I feel the authors should directly start with the problem of RSL with graph instead of formulating the (homophilous) graph as the ``side-information`` of the original RSL problem. Note that all the downstream evaluation heavily relies on the (supervised) performance of GCN. I feel it is misleading to merely term the graph as a side-information, as it seems to be the ``main source’’ of information for the problem.

If we view the studied problem as RSL with graph, then it is natural to relate it to the contextual stochastic model (cSBM) literature [1]. In fact, the GFoN model given by the authors in section 5 is very similar to cSBM, except that there is no prior distribution assumption (gaussian in cSBM) on the node features. Note that cSBM emerges as an important attributed random graph model for GNN analysis in both theory and practice [2-4], albeit they focus on (semi-)supervised learning setting. Nevertheless, the cSBM paper [1] itself and its follow-up study the community detection problem and information theoretical limit is established. I think the authors should discuss the relation of their work to cSBM and use cSBM algorithms (or even SBM algorithms) as baselines in the experiments. For example, simple spectral clustering (instead of MinCUT) with Kmeans.

Finally, I find the hardness results for RSL kind of **contradict** with the strong empirical performance of RS-GNN when the graph is induced from node features (Section 7.2). Note that in this case, the problem is exactly the RSL problem without any ``side-information``. Thus, the hardness result proved in Section 4 should still applied. In this point of view, I wonder how the RS-GNN can still perform well? Based on Theorem 4.2 (hardness result), there should be a hard example (as proposed by the author) that RS-GNN does not work in this case. I understand that RS-GNN without graphs may work well in practice but the flow of the paper is contradicting this fact. Also, I feel the authors should test RS-GNN without graphs on the hard example constructed in Section 4.


Minor: why use $m$ instead of $n$ to denote the number of nodes? I don’t see any reason for detouring from the standard graph learning literature.

Issue: The footnote on page 10 violates the double-blind policy of TMLR. Please make sure the paper is properly anonymized.

## References

[1] Contextual stochastic block models, Deshpande et al., NeurIPS 2018.

[2] Graph Convolution for Semi-Supervised Classification: Improved Linear Separability and Out-of-Distribution Generalization, Baranwal et al., ICML 2021.

[3] Effects of Graph Convolutions in Multi-layer Networks, Baranwal et al., ICML 2023.

[4] Adaptive universal generalized pagerank graph neural network, Chien et al., ICLR 2021.

---

> ### Author Response · Authors · 2023-06-13
> **Author Response**
>
> We thank the reviewer for their thoughtful comments. In our revision, we have made a number of changes that we hope will resolve the reviewer’s concerns (most changes are highlighted in blue for your convenience). Below are the answers to specific questions.
>
> > Lack of discussion and comparison to the relevant contextual stochastic block model literature.
>
> We included a discussion of the cSBM literature in section 5.
>
> > simple spectral clustering (instead of MinCUT) with Kmeans baseline
>
> We tested two versions of the proposed baseline and added the results to the paper in our revision (Tables 1 and 2).
>
> > The introduced side-information (homophilous graph) makes the RSL problem a different problem, and the hardness of this problem is unclear in this paper.
>
> The worst case analysis extends to the case of RSL with graphs, because in the worst case the graph can be random and provide no extra information. However, obtaining hardness results in the case where the graph has a certain degree of homophily is a much more difficult problem, and a solution to that problem may deserve an entire separate paper.
>
> > The hardness results kind of contradict the claim that RS-GNN can work without graphs (i.e., case of graphs induced from node features).
>
> We updated our claim about the results without graphs, to highlight that we expect the similarity graph structures to have some degree of homophily. This assumption is implicitly made in many previous works in the graph structure learning literature, including the (Fatemi et al. 2021) reference provided in the paper (see hypothesis 1 in their paper and its description).
>
> > I think the flow of the paper is problematic and misleading. Firstly, I feel the authors should directly start with the problem of RSL with graph instead of formulating the (homophilous) graph as the side-information of the original RSL problem.
>
> Upon your suggestion, we updated the structure of the paper so that we directly start with the RSL with graph (now termed GRSL in the paper – see Definition 3), and avoided calling the graph “side-information” throughout the paper.
>
> > Also, I feel the authors should test RS-GNN without graphs on the hard example constructed in Section 4.
>
> We tested RS-GNN and the baseline suggested by the reviewer on FoN with a similarity graph. As expected, both models performed close to random. This provides empirical evidence for the hardness of RSL in the worst case. We added this to section 7.2 of the paper.
>
> > Minor: why use m instead of n to denote the number of nodes? I don’t see any reason for detouring from the standard graph learning literature.
>
> We only did this because our complexity analysis depends on the number of features and we wanted to use a big-O notation which depends on n (following the convention). We are happy to switch n and m if the reviewer believes that makes the paper more clear.

---

> > ### Comment · Reviewer_Mgut · 2023-06-29
> > **Re:**
> >
> > Thank you for the detailed response. Most of my concerns are addressed.

---

### Review · Reviewer_STeE · 2023-05-31

**Summary Of Contributions:**

The paper tackles the problem of representative selection (RS), which is the problem of finding a small subset of exemplars from an unlabeled dataset, and has numerous applications in various domains. The paper focuses on finding representatives that optimize the accuracy of a model trained on the selected representatives. The paper studies RS for data represented as attributed graphs, where each node has a feature vector and a graph context specifying the connections between nodes. The paper develops RS-GNN, a representation learning-based RS model based on Graph Neural Networks (GNN). The paper demonstrates the effectiveness of RS-GNN on problems with predefined graph structures as well as problems with graphs induced from node feature similarities, by showing that RS-GNN achieves significant improvements over established baselines that optimize surrogate functions. The paper also establishes a new hardness result for RS by proving that RS is hard to approximate in polynomial time within any reasonable factor, which implies a significant gap between the optimum solution of widely-used surrogate functions and the actual accuracy of the model, and provides justification for the superiority of representation learning-based approaches such as RS-GNN over surrogate functions.

**Audience:**

Yes

**Claims And Evidence:**

Yes

**Requested Changes:**

Please kindly refer to Section Strengths and Weaknesses for detailed suggestions.

**Strengths And Weaknesses:**

Strengths:
* This manuscript studies an interesting problem, representative selection, which is of great importance from both theoretical and practical perspectives.
* The authors theoretically proved a hardness result showing that RS is hard to approximate in polynomial time within any reasonable factor under the exponential time hypothesis.
* The design of RS-GNN is simple yet effective. This paper further provides empirical evidence that shows the effectiveness of RS-GNN on various datasets and tasks, compared to existing methods.

Weaknesses:
* While I enjoyed reading this paper as it is almost clear everywhere, the related work section can be adjusted to be more concise. I agree that more discussion on related works could provide more background knowledge to the readers but the current version seems a little redundant, especially in comparison with the other parts of this manuscript. The authors might consider moving some of the discussions on the related works to the Appendix for a more concentrated demonstration.

* It would be better to introduce more on the ETH assumption and the background of the Fit-or-Not learning problem as well as their relationship to the RS problem in previous literature. This could make the theoretical findings better understood by audiences that are not from this research direction. Meanwhile, the connection between the Fit-or-Not learning problem and the used benchmark datasets is unclear. The authors are suggested to justify this relation by analysis of synthetic datasets or discussing examples of how the Fit-or-Not learning problem is usually employed in practical scenarios.

* The authors set the number to be $kc$ where $c$ is the number of classes from datasets. However, $c$ could be unknown in many applications. Could the authors provide more insights on how to select the number of representatives during practice? Potential sensitive analysis might help the authors to choose this hyperparameter.

---

> ### Author Response · Authors · 2023-06-13
> **Author Response**
>
> We thank the reviewer for their thoughtful comments. In our revision, we have made a number of changes that we hope will resolve the reviewer’s concerns (most changes are highlighted in blue for your convenience). Below are the answers to specific questions.
>
> > The related work section can be adjusted to be more concise.
>
> Upon your suggestion, we delegated the categories of related work that were more distant to our work to the Appendix. We believe the related work section is now more focused around the main bodies of work that relate to our work. Please let us know if any further change is needed..
>
> > It would be better to introduce more on the ETH assumption and the background of the Fit-or-Not learning problem as well as their relationship to the RS problem in previous literature.
>
> We added an introduction and a more detailed description of the ETH assumption in Section 4 of the paper.
>
> > Meanwhile, the connection between the Fit-or-Not learning problem and the used benchmark datasets is unclear. The authors are suggested to justify this relation by analysis of synthetic datasets or discussing examples of how the Fit-or-Not learning problem is usually employed in practical scenarios.
>
> FoN is a basic artificial learning problem that we defined to show that there exists a hard RSL problem, and hence theoretically RSL in its most general form can not be solved in polynomial time. In practice (including for our datasets), the problems may be at different levels of difficulty, but not necessarily at a worst-case. We added an example scenario where elements of the FoN problem can appear in real-world problems in Section 4.
>
> > The authors set the number to be kc where c is the number of classes from datasets. However, c could be unknown in many applications. Could the authors provide more insights on how to select the number of representatives during practice?
>
> We set the value to kc so that we measure the performance of RSGNN and compare against baseline with various number of representatives selected across different datasets. We have provided experiments with k=2 in the main paper, and k=5 in the appendix. In practice, the number of examples to label is decided based on the budget one has for labeling data, or based on the coverage metrics discuss in the related work section.

---

> > ### Comment · Reviewer_STeE · 2023-06-25
> > **Thank you for the response**
> >
> > Thank you for the detailed and thoughtful feedback. I have also read the comments from other reviewers as well as the corresponding replies. My concerns are mostly addressed.

---

### Review · Reviewer_FGHE · 2023-06-08

**Summary Of Contributions:**

This paper considers the problem of selecting representative examples from a dataset such that training a classifier on the representatives will result in a good classifier. First, the authors prove a hardness result by reduction to a special case of this learning problem to densest $k$-subgraph problem. This means that any polynomial time algorithm scales poorly in the number of features unless the Exponential-Time Hypothesis fails. Then, the paper proposes a variant of the problem in which labeled data is unavailable, but noisy graph information reveals whether two examples are likely to have the same label. It then proposes RS-GNN, a method which jointly selects representatives while learning an intermediate representation based on Deep Graph Infomax. Experimental results show that RS-GNN performs well on transductive node classification for standard GNN datasets.

**Audience:**

Yes

**Broader Impact Concerns:**

Broader Impact statement sufficiently describes concerns about fairness across subpopulations.

**Claims And Evidence:**

Yes

**Requested Changes:**

### Critical
- Clearly explain that $n$ is the number of features (dimension) and is unrelated to the number of candidate points $m$ or number of selected points $k$, and how this differs from related work.
- Compute the hardness of approximation for the FoN instance shown in Figure 1.
- Add more details/ablation about the optimization in Algorithm 1: learning rate parameters, joint vs alternating minimization.
- In Section 7.2 mention that $k=15$ in the kNN graph (this important detail is in Appendix D).
- Comment on whether other representation learning baselines can utilize CenterNorm, and run additional ablations if applicable.
- Comment on drop in performance when evaluating with NMI instead of accuracy, and consider other evaluation metrics such as F1 score which better account for class imbalance. Are error bars missing from Table 7 because the results are from another paper?


### Non-critical
- Varying $k$ as a function of number of labels/clusters $c$ with each dataset makes sense. However, the authors should comment on what can be done when $c$ is not known in advance. Also it would be good to consider a higher number of representatives in the range $5c < k < \min \\{m, n\\}$.
- Mention additional related work on joint clustered representation learning: VQ-VAE [1] and a graph-based variant VQ-GNN [2].
- Discuss/consider an appropriate regularization penalty in addition to CenterNorm and ConstNorm in Table 5.
- Hardness results consider a problem instance where $m = 2n$. It would also be interesting to consider the case where $m \gg n$ since this is common for many applications (including several datasets in Section 7).
- Consider running experiments with fixed train/validation/test sets, and selecting representatives from the training set $\mathcal{V}$. This is still a transductive learning problem, but every algorithm will be predicting on the same holdout set instead of $\mathcal{V} \setminus \mathcal{S}$.
- Figure 3(a) is less useful without comparing to baselines as in Figure 4(a)-(b). If there is not enough space, then Figure 3(a) can be moved to the Appendix.


### Typos
- Definition 3.2: $y$ should be $Y$
- Section 6.2: "have prove effective" should be "have proved effective"
- Table 4: colors are missing
- Appendix D: "Therefore, we For the" should be "Therefore, for the"

[1] van den Oord et al. [Neural Discrete Representation Learning](https://arxiv.org/abs/1711.00937), NeurIPS 2017.

[2] Ding et al. [VQ-GNN: A Universal Framework to Scale up Graph Neural Networks using Vector Quantization](https://arxiv.org/abs/2110.14363), NeurIPS 2021.


**Strengths And Weaknesses:**

### Strengths
- Interesting problem formulation which combines several related subproblems.
- Theoretical results appear to be novel and correct.
- Proposed algorithm is simple and practically motivated.
- Good comparison to relevant baselines.

### Weaknesses
- Organization: The hardness result is based on the FoN Learning Problem, while the proposed RS-GNN is based on GFoN version of the problem. The theory and empirical contributions are only loosely connected because there are no theoretical results for the setting of the proposed method. Showing empirical performance of RS-GNN as a function of feature dimension might improve this.
- Methodology:
	- Figure 4 shows that CenterNorm is quite important to the performance of the full model. Experiments would be more complete if they showed how CenterNorm can be applied to other representation learning methods (e.g. EGAE, GCC).
	- Visualizations use t-SNE in Figure 2 and UMAP in Figures 3-4. This should be explained or made consistent.
- Results:
	- Graph-based evaluation (NMI) is less consistent/impressive than evaluating using Accuracy.
	- In addition to large class imbalance, it is also worth noting that Arxiv has the largest number of nodes/edges/classes and lowest number of base features.
- Soundness: Previous work on example selection typically assumes the feature dimension $n$ is constant. This should be made very clear whenever the hardness result is compared to the constant $1 - 1/e$ approximation ratio, e.g. before Corollary 4.3.
- Some missing optimization details (see below).

---

> ### Author Response · Authors · 2023-06-13
> **Author Response**
>
> We thank the reviewer for their thoughtful comments. In our revision, we have made a number of changes that we hope will resolve the reviewer’s concerns (most changes are highlighted in blue for your convenience). Below are the answers to specific questions.
>
> > Clearly explain that n is the number of features (dimension) and is unrelated to the number of candidate points m or number of selected points k, and how this differs from related work.
>
> We added a clear explanation in section 3 following your suggestion.
>
> > Compute the hardness of approximation for the FoN instance shown in Figure 1.
>
> Please note that the hardness result that we provide talks about the asymptotic running time (a.k.a.  the Order of the running time). Hence, it does not apply to a small example with constant size.
>
> > Add more details/ablation about the optimization in Algorithm 1: learning rate parameters, joint vs alternating minimization.
>
> Added to Appendix D (implementation details).
>
> > In Section 7.2 mention that k=15  in the kNN graph (this important detail is in Appendix D).
>
> Added.
>
> > Comment on whether other representation learning baselines can utilize CenterNorm, and run additional ablations if applicable.
>
> The central idea behind CenterNorm is optimized for DGI-style models, because of their latent space property described in Appendix B. Other representation learning models may require their own normalization schemes that match better with their latent space properties. In other words, our normalization approach provides a general recipe that other models can leverage by specilizing it to their latent space property.
>
> > Comment on drop in performance when evaluating with NMI instead of accuracy.
>
> Performing well in terms of NMI mainly requires being able to properly cluster the nodes, whereas the accuracy metric in our work requires both properly clustering the nodes and also selecting good cluster representatives. The higher gap between RSGNN and existing work in terms of accuracy compared to NMI shows that while existing work might be more competitive in terms of node clustering, their performance is substantially lower than RSGNN in terms of selecting good representatives for the clusters. Note that this is expected as previous work has mainly optimized for NMI, and ignores the representative selection part.
>
> > Varying k as a function of number of labels/clusters c with each dataset makes sense. However, the authors should comment on what can be done when c is not known in advance.
>
> We set the value based on the number of labels/clusters so that we measure the performance of RSGNN and compare against baseline with various number of representatives selected across different datasets. In practice, the number of examples to label is decided based on the budget one has for labeling data, or based on the coverage metrics discuss in the related work section.
>
> > Mention additional related work on joint clustered representation learning: VQ-VAE [1] and a graph-based variant VQ-GNN [2].
>
> We added references in our revision.
>
> > Hardness results consider a problem instance where m=2n. It would also be interesting to consider the case where m >> n since this is common for many applications (including several datasets in Section 7).
>
> We clarify that the instance we present in Figure 1 is only for the sake of understanding the Fit or Not problem; nothing about our theorem depends on that instance or the specifics of it (in particular, we are not assuming m=2n).
>
> > Are error bars missing from Table 7 because the results are from another paper?
>
> Yes. The results come from the DMoN paper.
>
> > In addition to large class imbalance, it is also worth noting that Arxiv has the largest number of nodes/edges/classes and lowest number of base features.
>
> Added.
>
> > Visualizations use t-SNE in Figure 2 and UMAP in Figures 3-4.
>
> We used UMAP for Figures 3-4 only because the cluster boundaries were slightly more clear than t-SNE plots. Switching back to t-SNE provides similar conclusions.
>
> > Previous work on example selection typically assumes the feature dimension n  is constant.
>
> This is now added in the revision.
>
> > Typos
>
> Thanks, the typos are now fixed. (The color-coding in Table 4 is different from Tables 1 and 2: We use the blue color to represent the baselines that use DGI embeddings).

---

> > ### Comment · Reviewer_FGHE · 2023-06-30
> >
> > The author response has addressed most of my concerns, and the paper is stronger now.

---

### Author Response · Authors · 2023-07-17
**Final Decision**

We would like to thank the reviewers and the action editor one more time for their time and valuable feedback. The review process was immensely helpful and improved our submission in many ways. Based on the final reviewer comments in the discussion period, it seems like the concerns raised by the reviewers have been resolved. While we wait for the AE's final decision, we are happy to discuss and address any other concerns the reviewers or the AE might have.

---

### Decision · Action_Editors · 2023-07-18

**Recommendation:** Accept with minor revision

**Comment:**

See above.

**Audience:**

Yes, the paper is interesting for TMLR's audience.

**Claims And Evidence:**

The paper studies the data subset selection problem for attributed graphs. The contributions are twofold: a general hardness result and a particular feasibility result in the presence of graph structure. The paper gives clear evidence to support these claims. All the reviewers have favored acceptance. I agree with this assessment.

Minor comment:
In begin of Section 4, there is a misplaced sentence "Needs an opening sentence relating RSL to GRSL!". Please remove that.
An initial common comment was on the readability of the paper on how rsl hardness and grsl discussion seemed disparate. As a follow-up, the current revision has addressed the concerns. However, I would still encourage the authors to polish the paper if needed.